

# Challenges issues and future recommendations of deep learning techniques for SARS-CoV-2 detection utilising X-ray and CT images: a comprehensive review

Md Shofiqul Islam[1,2], Fahmid Al Farid[3], F. M. Javed Mehedi Shamrat[4], Md Nahidul Islam[5], Mamunur Rashid[5,6], Bifta Sama Bari[5,6], Junaidi Abdullah[7], Muhammad Nazrul Islam[1], Md Akhtaruzzaman[1], Muhammad Nomani Kabir[8], Sarina Mansor[3] and Hezerul Abdul Karim[3]

[1] Computer Science and Engineering (CSE), Military Institute of Science and Technology (MIST), Dhaka, Bangladesh
[2] Institute for Intelligent Systems Research and Innovation (IISRI), Deakin University, Warun Ponds, Victoria, Australia
[3] Faculty of Engineering, Multimedia University, Cyeberjaya, Selangor, Malaysia
[4] Department of Computer System and Technology, Universiti Malaya, Kuala Lumpur, Malaysia
[5] Faculty of Electrical and Electronics Engineering Technology, Universiti Malaysia Pahang Al-Sultan Abdullah (UMPSA), Pekan, Pahang, Malaysia
[6] Electrical and Computer Engineering, Tennessee Tech University, Cookeville, TN, United States
[7] Faculty of Computing and Informatics, Multimedia University, Cyberjaya, Selangor, Malaysia
[8] Department of Computer Science & Engineering, United International University (UIU), Dhaka, Bangladesh

Corresponding authors
Fahmid Al Farid,
fahmid.farid@gmail.com
Hezerul Abdul Karim,
hezerul@mmu.edu.my

## ABSTRACT

The global spread of SARS-CoV-2 has prompted a crucial need for accurate medical diagnosis, particularly in the respiratory system. Current diagnostic methods heavily rely on imaging techniques like CT scans and X-rays, but identifying SARS-CoV-2 in these images proves to be challenging and time-consuming. In this context, artificial intelligence (AI) models, specifically deep learning (DL) networks, emerge as a promising solution in medical image analysis. This article provides a meticulous and comprehensive review of imaging-based SARS-CoV-2 diagnosis using deep learning techniques up to May 2024. This article starts with an overview of imaging-based SARS-CoV-2 diagnosis, covering the basic steps of deep learning-based SARS-CoV-2 diagnosis, SARS-CoV-2 data sources, data pre-processing methods, the taxonomy of deep learning techniques, findings, research gaps and performance evaluation. We also focus on addressing current privacy issues, limitations, and challenges in the realm of SARS-CoV-2 diagnosis. According to the taxonomy, each deep learning model is discussed, encompassing its core functionality and a critical assessment of its suitability for imaging-based SARS-CoV-2 detection. A comparative analysis is included by summarizing all relevant studies to provide an overall visualization. Considering the challenges of identifying the best deep-learning model for imaging-based SARS-CoV-2 detection, the article conducts an experiment with twelve contemporary deep-learning techniques. The experimental result shows that the MobileNetV3 model outperforms other deep learning models with an accuracy of 98.11%. Finally, the article elaborates on the current challenges in deep

learning-based SARS-CoV-2 diagnosis and explores potential future directions and methodological recommendations for research and advancement.

# INTRODUCTION

The second subtype of coronavirus, known as SARS-CoV-2 (severe acute respiratory syndrome coronavirus 2), is a viral entity accountable for causing respiratory diseases in humans (*Magno et al., 2020*). In certain countries, the emergence of coronavirus disease 2019 (SARS-CoV-2) has had detrimental effects on healthcare systems. The surge in patients exhibiting severe SARS-CoV-2 symptoms has strained both public and private healthcare facilities, resulting in a significant rise in the number of affected individuals. Consequently, several governments have instituted large-scale testing initiatives to facilitate early diagnosis and mitigate the strain on healthcare systems. To confirm the presence of SARS-CoV-2, the real-time polymerase chain reaction (RT-PCR) method, performed using combined oral/nasal swabs, detects the virus's genetic material in lung samples (*Magno et al., 2020*; *Teodoro et al., 2021*). However, there are inherent challenges associated with this testing approach. The virus's genetic material can be detectable within a few days of symptom onset but may become undetectable after a fourteen-day period (*Magno et al., 2020*). Furthermore, the complexity of the test itself presents obstacles, exacerbated by the lack of necessary infrastructure and technical proficiency in many healthcare centers (*Magno et al., 2020*; *Barbosa et al., 2020*; *Hu, Qiu & Zhou, 2022*).

Various artificial intelligence approaches have been effectively employed in a variety of medical research domains (*Bao et al., 2021*; *Lu et al., 2024*; *Hu et al., 2024*). In terms of precision and performance in image categorization, the deep learning-based methodology beats conventional methods (*Khandokar et al., 2021*; *Sitaula & Hossain, 2021*; *Ozyurt, Tuncer & Subasi, 2021*; *Zebin & Rezvy, 2021*; *Ahsan et al., 2021*; *Ardakani et al., 2020*; *Al Farid et al., 2022*). Various machine learning techniques are used to classify images or detect and describe specific regions of interest in the examined images, particularly medical images and videos have been embraced as a useful source of information for diagnosing purposes. Various approaches are also employed to increase image quality (*Singh et al., 2021*; *Chen et al., 2020*; *Zheng et al., 2024*). A denoising filter based on local statistics is built in *Singh et al. (2020)* to increase image quality, which is vital for delivering an accurate medical diagnosis of a condition. Because pneumonia develops in extreme situations of SARS-CoV-2, the use of radiographic medical images, including chest radiography (X-ray), chest ultrasonography, and chest computerized tomography (CT), has been used to aid in the early diagnosis of SARS-CoV-2 and also to rule out the suspicion of illnesses (*Barbosa et al., 2020*; *Wang et al., 2021c*). The experimental results achieved using CT pictures were superior to those obtained using x-ray images, according to *Wang et al. (2021c)*. It is also worth noting that various chest CT datasets are frequently utilised for research and have

recently been employed in the diagnosis of SARS-CoV-2. Chest CT images have been seen in certain investigations to indicate signs of SARS-CoV-2 development in patients, such as bilateral alterations with ground-glass opacity even without consolidating and interlobular septal thickness (*Teodoro et al., 2021*; *Magno et al., 2020*).

Machine learning, as well as deep learning (DL), are intriguing technologies that many healthcare professionals are using since they allow humans to perform actions better and faster. As a consequence, to tackle the SARS-CoV-2 problem (*Nugraheni et al., 2020*; *Dastider, Sadik & Fattah, 2021*; *Jain et al., 2021*; *Sitaula & Hossain, 2021*), the medical industry has adopted a range of deep learning technologies all around the world. As a result, healthcare companies and physicians all over the world deployed a variety of deep learning technologies to combat the SARS-CoV-2 pandemic and overcome the barriers that arose during the epidemic. Instead of replacing human knowledge, deep learning is employed in the medical business to provide choice assistance for clinicians on what they are modelled for *Phillips-Wren & Ichalkaranje (2008)*. Deep learning ensures better precision (*Lella & Pja, 2022*; *Sait et al., 2021*; *Pinkas et al., 2020*) and versatility (*Lella & Pja, 2022*; *Paka et al., 2021*).

Some classic symptoms of the novel coronavirus, such as cough and fever, have been observed and detected. However, in certain cases, there are no signs (symptomless). The most commonly observed and detected symptoms of the coronavirus are cough and fever, *etc* (*Phillips-Wren & Ichalkaranje, 2008*; *Davenport & Kalakota, 2019*). However, in certain cases, no symptoms are apparent (asymptomatic), making the illness much more dangerous to public health.

Medical imaging analysis offers numerous advantages over sequencing-based COVID-19 analysis (*Hemdan, Shouman & Karar, 2020*; *Sitaula & Hossain, 2021*). Firstly, its deep learning algorithms process chest X-rays and CT scans swiftly, yielding results within minutes, crucial for prompt clinical decision-making, especially during pandemics. Secondly, being non-invasive, enhances patient compliance and comfort compared to invasive swab tests, thus facilitating initial screenings and routine monitoring. Moreover, it provides detailed visualization of COVID-19-related pathologies, aiding in precise diagnosis. Leveraging existing infrastructure, optimizes resource allocation and reduces financial burden, particularly in resource-limited settings. Additionally, it enables dynamic disease progression monitoring, empowering healthcare professionals to make informed treatment adjustments. In summary, medical imaging analysis proves invaluable in COVID-19 diagnosis and management, offering unparalleled speed, non-invasiveness, visualization, resource optimization, and monitoring capabilities.

Image-based (*Jain et al., 2021*; *Ozyurt, Tuncer & Subasi, 2021*; *Zebin & Rezvy, 2021*; *Ahsan et al., 2021*; *Ardakani et al., 2020*; *Hemdan, Shouman & Karar, 2020*) methods gives more precession in detection of SARS-CoV-2 (*Ardakani et al., 2020*; *Lella & Pja, 2022*; *Paka et al., 2021*). When available, different imaging techniques, including CT and X-ray, are considered to be among the most effective tactics for SARS-CoV-2 diagnosis (*Ardakani et al., 2020*). Image acquisition modalities, such as CT as well as X-ray, are thought as one of the most significant techniques for SARS-CoV-2 diagnosis (*He et al., 2020*). Under which CT scanning seems to be available, it is preferred over X-rays because of its

simplicity as well as 3D pulmonary (*Ardakani et al., 2020*; *Kim et al., 2020*) view, despite the fact that X-rays are more available as well as widely available. These traditional medical imaging technologies are important in the pandemic's containment (*Kim et al., 2020*).

Artificial Intelligence (AI), a fast-developing computing technology in medical image analysis, has also benefited in the fight against new coronaviruses (*Kim et al., 2020*) by efficiently achieving great diagnostic outputs while radically reducing or eliminating the need for human interaction (*Jiang et al., 2022*). Deep learning as well as machine learning, two major branches of AI, have lately acquired traction in medical applications. Image (CT and X-ray) (*Kim et al., 2020*; *Ardakani et al., 2020*; *Lella & Pja, 2022*; *Paka et al., 2021*) samples are being used to build deep learning-based support mechanisms for SARS-CoV-2 diagnosis. Some of the systems are based on transfer learning models that have already been pre-trained (*Dourado et al., 2019*), while others are based on individualised networks (*Ozturk et al., 2020*).

## Relevant review articles

This article explores the domain of deep learning techniques used for analyzing SARS-CoV-2. Within the realm of deep learning-based analysis for SARS-CoV-2, numerous reviews and survey publications are available. Table 1 outlines recent deep learning-based review articles centred around SARS-CoV-2 analysis. A review article spotlighted the latest advancements in AI methodologies for detecting SARS-CoV-2 using chest radiography and computed tomography; however, it falls short in addressing limitations and research gaps (*Aslani & Jacob, 2023*). Another recent survey showcased the development of deep learning techniques, comprehensively illustrating diverse methodologies and their performance. Yet, it does not delve into research gaps (*Subramanian et al., 2022*). In the year 2023, an article (*Ait Nasser & Akhloufi, 2023*) presented a review detailing 22 publicly available datasets employed for automatic chest disease detection using radiological medical images and DL techniques. It elucidated preprocessing and data-augmentation techniques, engaged with research challenges, and explored alternatives. Nonetheless, this work does not explicitly delineate key findings and intricate mechanisms. A review article discussed the gamut of deep learning models for coronavirus (SARS-CoV-2) epidemic analysis with technology and application. Regrettably, it lacks a robust methodological exposition (*Awassa et al., 2022*). *Shorten, Khoshgoftaar & Furht (2021)* contributed a novel review study on SARS-CoV-2 analysis employing deep learning. This article explored how artificial neural networks feed data to deep neural networks, construct learning paradigms, tackle tasks, and analyze novel datasets, including the utilization of annotation techniques. However, this review fell short in adequately addressing recent limitations, research gaps, findings, and discoveries. *Syeda et al. (2021)* presented another review, examining novel machine learning techniques in SARS-CoV-2 investigation, scrutinizing recent literature that employed AI techniques to trace, comprehend, and combat virus inflammation. Despite its contribution, this work lacks comprehensive methodological descriptions and research gap identification. *Sharma & Guleria (2024)* contributed a review article categorically summarizing findings, yet it did not extensively explore research gaps. A review study furnished a decent examination of recent methodologies for SARS-CoV-2

**Table 1 Review characteristics.**

| Review features | Details of the review article features |
|---|---|
| C1 | Background analysis with relevant existing study on focused application. |
| C2 | Finding limitation of existing review and recovery idea to solve some of the limitations. |
| C3 | Cover data information and its reprocessing to get good result of the deep learning method. |
| C4 | Clearly describe the feature extraction process before it is sent to the deep learning model. |
| C5 | Ensure types of method with applications for SARS-CoV-2 analysis |
| C6 | Describe method in details with algorithms and figures |
| C7 | State out all the classification methods based on recent applications |
| C8 | State out multi-modal review to cover all field deep learning method |
| C9 | Critical result analysis on recent methods to give deeper analysis with comparison. |
| C10 | Mentioning of research gaps of existing system to motivate researcher |
| C11 | Giving overall summary with mentioning future direction to understand trends and directions |
| C12 | Covering challenges and limitations that encourage researchers to develop new approach |
| C13 | Providing recommendation based on comparative analysis |
| C14 | Ensuring recommendation based on the analytical analysis performance of the existing methods. |
| C15 | Covering latest motivations to develop more successful method in SARS-CoV-2 detection. |

| Reference | Review features addressed | | | | | | | | | | | | | | |
|---|---|---|---|---|---|---|---|---|---|---|---|---|---|---|---|
| Author and year | C1 | C2 | C3 | C4 | C5 | C6 | C7 | C8 | C9 | C10 | C11 | C12 | C13 | C14 | C15 |
| *Aslani & Jacob (2023)* | Y | Y | Y | Y | Y | Y | Y | | | | | | | | |
| *Ait Nasser & Akhloufi (2023)* | Y | Y | | Y | Y | Y | Y | | | | Y | Y | | | Y |
| *Subramanian et al. (2022)* | Y | Y | | Y | Y | Y | Y | | Y | | Y | Y | | | Y |
| *Awassa et al. (2022)* | Y | Y | Y | Y | Y | Y | Y | | Y | | Y | | | | Y |
| *Shorten, Khoshgoftaar & Furht (2021)* | Y | Y | Y | Y | Y | Y | Y | | Y | | Y | | | | Y |
| *Syeda et al. (2021)* | Y | Y | Y | Y | Y | | Y | | Y | | Y | | | | |
| *Islam et al. (2020)* | Y | Y | | Y | Y | Y | Y | | Y | | Y | | | | |
| *Islam, Islam & Asraf (2020)* | Y | Y | Y | Y | Y | | Y | | Y | | Y | | | | |
| *Deshpande & Schuller (2020)* | Y | Y | Y | Y | Y | Y | Y | Y | Y | | Y | | | | |
| *Manigandan et al. (2020)* | Y | Y | | Y | Y | Y | Y | | Y | | Y | | | | |
| Ours | Y | Y | Y | Y | Y | Y | Y | | Y | Y | Y | Y | Y | Y | Y |

analysis, albeit methodological explanations were lacking (*Subramaniam et al., 2023*). *Manigandan et al. (2020)* compiled an inclusive review encompassing recent trends in SARS-CoV-2 transmission, diagnosis, prevention, and imaging features. Regrettably, this publication did not sufficiently elucidate research gaps and methodological intricacies. Another review expounded on the use of deep learning techniques in analyzing SARS-CoV-2 within clinical practice, specifically employing CT images. However, it only examined the results of 10 convolutional neural networks and outlined 10 approaches without addressing gaps in the literature (*Ardakani et al., 2020*). A review study centered on SARS-CoV-2 estimation and forecasting provided a succinct overview, yet it did not accentuate the drawbacks of each method and focused primarily on the outcomes of seven models (*Salehi, Baglat & Gupta, 2020*).

By and large, many of these studies employ traditional methods to dissect contemporary SARS-CoV-2 research. Existing survey articles often provide concise explanations to illustrate methodologies and related works in a generalized manner. A recurring pattern is observed in traditional survey articles for SARS-CoV-2 research, frequently encompassing the description of the compared samples or related studies.

There are a few essential features to look for when judging a successful review article. We have compiled a list of the top fifteen characteristics that guarantee a high-quality review article. If any literature reviews fail to report a few of the review system's characteristics, then such articles suggest that some aspects of the review system are missing.

Table 1 shows the limitations and findings of the existing review based on our selected fifteen characteristics of a good review article.

Our comprehensive survey article thoroughly investigates the SARS-CoV-2 study's classification system, prevailing architecture, structures, and dataset utilization. This encompasses a succinct overview of data attributes, challenges, performance metrics, output evaluations, and prominent recent research citations, shedding light on the foundational equations that underlie deep learning algorithms.

In the critical review segment of our report, we expound on analogous research endeavours, providing specific insights into their methodologies, datasets, the quantity of class predictions, performance results, findings, and areas of research gaps. This critical assessment spans the entire spectrum of deep learning-based SARS-CoV-2 analytical techniques, offering a concise summary based on specified attributes.

Our main contribution to this review study can be presented as follows:

- To investigate systematically on state-of-the-art deep learning breakthroughs on imaging models in SARS-CoV-2 medical diagnostics, including categorization, recent applications, methodology, data, and obstacles.
- Discuss the overall findings, performances, uses, and research gaps, together with pertinent recent research references, basic datatype operations, deep learning models with fully illustrated structure and formulas, and the SARS-CoV-2 evaluation algorithm.
- Critical comparison of all modern SARS-CoV-2 analysis methods based on deep neural networks including advantages, weaknesses, limitations, research directions, and concern about some of the most successful methods.
- Experimentally analyze the twelve most sophisticated deep learning techniques for SARS-CoV-2 analysis, with a focus on the most useful technique.
- To discuss the challenges, privacy, benefits and drawbacks of deep learning models and recommend the best deep learning tools for successful SARS-CoV-2 detection.

The arrangement of the subsequent sections in this study is detailed in Fig. 1. The initial segment furnishes background information, while "Overview of Deep Learning Approaches for SARS-CoV-2 Analysis" furnishes a step-by-step synopsis of deep learning models. Expounding upon various deep learning techniques for SARS-CoV-2 detection, "Deep Learning Methods" offers a methodological perspective. In "Result Analysis", a

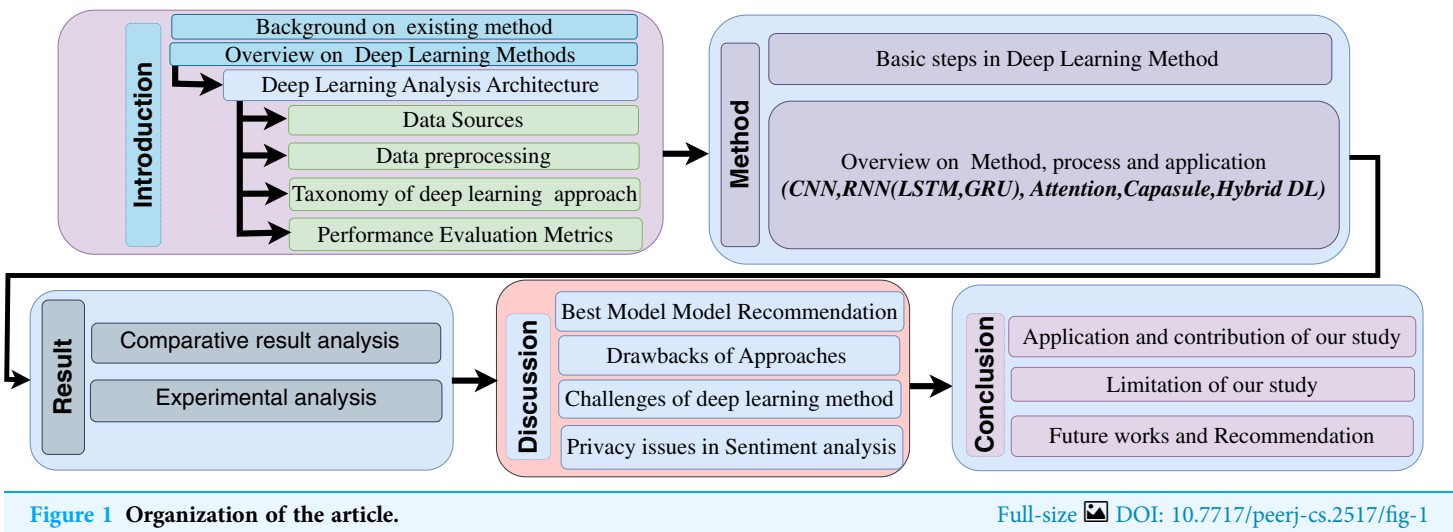

**Figure 1 Organization of the article.**

thorough exploration of recent research is presented, encompassing methodologies, approaches, data employment, outcomes, and areas of research lacunae.

Furthermore, "Discussion and Recommendation" encapsulates the analysis of results, entailing comparative and experimental assessments, ultimately culminating in the recommendation of the optimal model for COVID-19 detection. "Conclusion" is dedicated to delimitations, privacy considerations, challenges, and suggestions for forthcoming endeavours.

## Rationale and intended audience of the research

Amid the SARS-CoV-2 pandemic, a thorough examination of deep learning methods for detecting the virus using X-ray and CT images is essential. Medical imaging plays a crucial role in spotting SARS-CoV-2-related complications, yet obstacles such as data scarcity and interpretability issues impede advancements. This review offers a guide for researchers and policymakers to improve diagnostic precision and patient results. By tackling present constraints and directing future investigations, it encourages innovation and cooperation in crafting efficient diagnostic instruments to combat SARS-CoV-2.

This comprehensive review is tailored for researchers and academics deeply engaged in the development and enhancement of deep learning techniques for COVID-19 detection through X-ray and CT images. Healthcare professionals, including radiologists and clinicians, seeking insights into the challenges and advancements in utilizing deep learning for diagnosing COVID-19, will find this review invaluable. Public health officials and policymakers involved in shaping strategies for COVID-19 management and pandemic response efforts can benefit from the recommendations outlined in this review. Industry stakeholders, such as medical imaging technology companies and AI developers, can gain valuable insights into collaboration opportunities and product development avenues in the field of deep learning-based COVID-19 diagnostics. Students and educators in medical imaging, machine learning, and public health disciplines will find this review a valuable

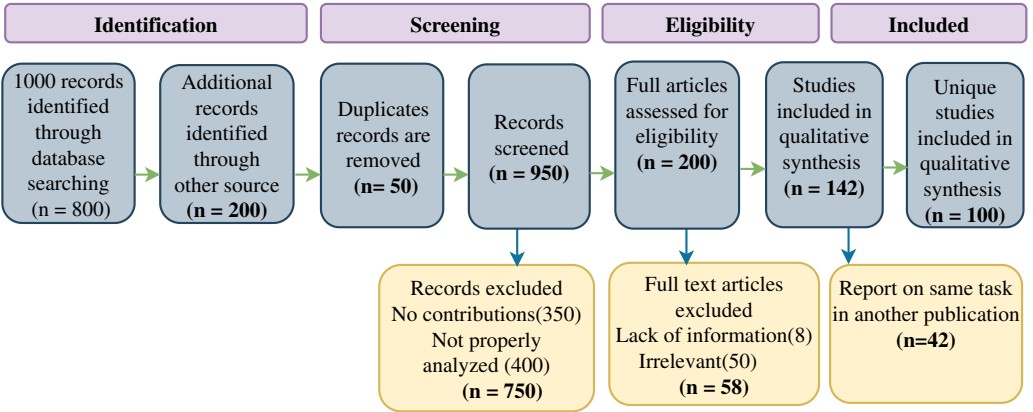

**Figure 2  Flow diagram of literature search for including studies in our systematic review.**

educational resource to understand the current landscape and future directions of COVID-19 detection using deep learning techniques.

## Article selection process

The process of article selection is elucidated in this section. Figure 2 presents a visual representation of the literature screening procedure, depicting the sequential inclusion and exclusion stages. The chosen articles stem from diverse reputable journals and were acquired through an internet database search. Electronic journals such as Scopus, IEEE Access, IEEE Explore, IEEE Transactions, Springer, ACM Library, Elsevier, SAGE, MDPI, Taylor and Francis, Wiley, Peerj, JSTOR, DBLP, and DOAJ were utilized for this purpose.

Certain articles were omitted during the study period due to specific reasons outlined in the accompanying diagram. The refinement of the final article selection transpired at both the screening and eligibility tiers. Initially, our analytical search yielded a total of 1,000 pertinent publications, from which 50 duplicate articles were eliminated. Among these articles, 750 redundant citations were excluded, leaving 200 for further examination. Applying the established criteria, 58 articles were subsequently excluded, including comprehensive reviews and review reports, while focusing exclusively on comprehensive articles that engaged in individual SARS-CoV-2 analysis.

Ultimately, the full texts of the remaining publications were thoroughly assessed, leading to the removal of 42 articles that had already been utilized in another publication addressing the same task. Consequently, our review analysis encompassed 100 manuscripts that met the stipulated inclusion criteria.

## OVERVIEW OF DEEP LEARNING APPROACHES FOR SARS-COV-2 ANALYSIS

The utilization of deep learning in SARS-CoV-2 analysis adheres to a structured procedure. It commences with the phase of data collection, which involves the aggregation of diverse medical images encompassing SARS-CoV-2 cases, other respiratory ailments, and healthy samples. Subsequent to data preprocessing, measures are undertaken to ensure

uniformity and eliminate disturbances by means of cleansing, standardization, and resizing. Data augmentation is then employed to expand the dataset's size through transformative techniques, thereby enhancing the model's ability to generalize. The phase of model selection entails the careful choice of a suitable deep learning architecture or model tailored to the specific analysis objective. This selection is succeeded by model training, during which the chosen model assimilates pivotal attributes for SARS-CoV-2 detection from the preprocessed dataset.

Figure 3 illustrates the recommended tools in the context of deep learning-based SARS-CoV-2 detection. The colour coding within the figure serves as an indicator of the tools' respective utility and significance.

## Open source data for SARS-CoV-2 analysis

This section delves into pertinent data concerning the analysis of SARS-CoV-2. Multiple common datasets are at the disposal of researchers engaged in SARS-CoV-2 detection and analysis. In this comprehensive review, our focus is on medical data, particularly two types of imaging data: CT scans and X-ray images sourced from medical repositories. These datasets originate from both public and private sources, as depicted in Fig. 4.

A succinct visualization of datasets utilized by various researchers in their SARS-CoV-2 analysis endeavours, detailing dataset types and quantities, is encapsulated in Fig. 5. This figure provides an informative overview of the dataset landscape corresponding to each model employed for SARS-CoV-2 analysis using image data. The color-coded bars denote different disease classes, while the X-axis signifies the number of data points harnessed for each model. The respective color keys for each class are conveniently provided on the right-hand side of the diagram.

Image-centric data is accessible from various repositories, with GitHub and Kaggle emerging as prime sources for freely available image datasets suitable for analysis. Predominantly, X-ray and CT scan images constitute the prevalent imaging modalities. Each of these categories possesses distinctive attributes, advantages, and limitations within the realm of research. X-ray image data offers ease of analysis and ready availability, albeit with drawbacks like the inability to detect certain conditions and manage loss of GGO density-based images. Conversely, CT scan images exhibit merits such as lower costs, heightened sensitivity, and greater reproducibility, but are accompanied by constraints such as limited accessibility and the need for multiple scans to ensure precision. Leveraging medical image processing technology could significantly contribute to SARS-CoV-2 analysis.

Furthermore, Table 2 enumerates publicly available data sources along with their corresponding links, data volumes, types, and class counts. This compilation facilitates researchers' access to well-established and widely used datasets.

In Fig. 6, we graphically depict the count of methods subjected to analysis within our article (part a), and in part b, we present the percentage distribution of deep learning applications covered in our comprehensive review.

Several publications have presented biometric analyses of SARS-CoV-2-related experiments. Notably, the Allen Institute for AI spearheaded the CORD-19 project in

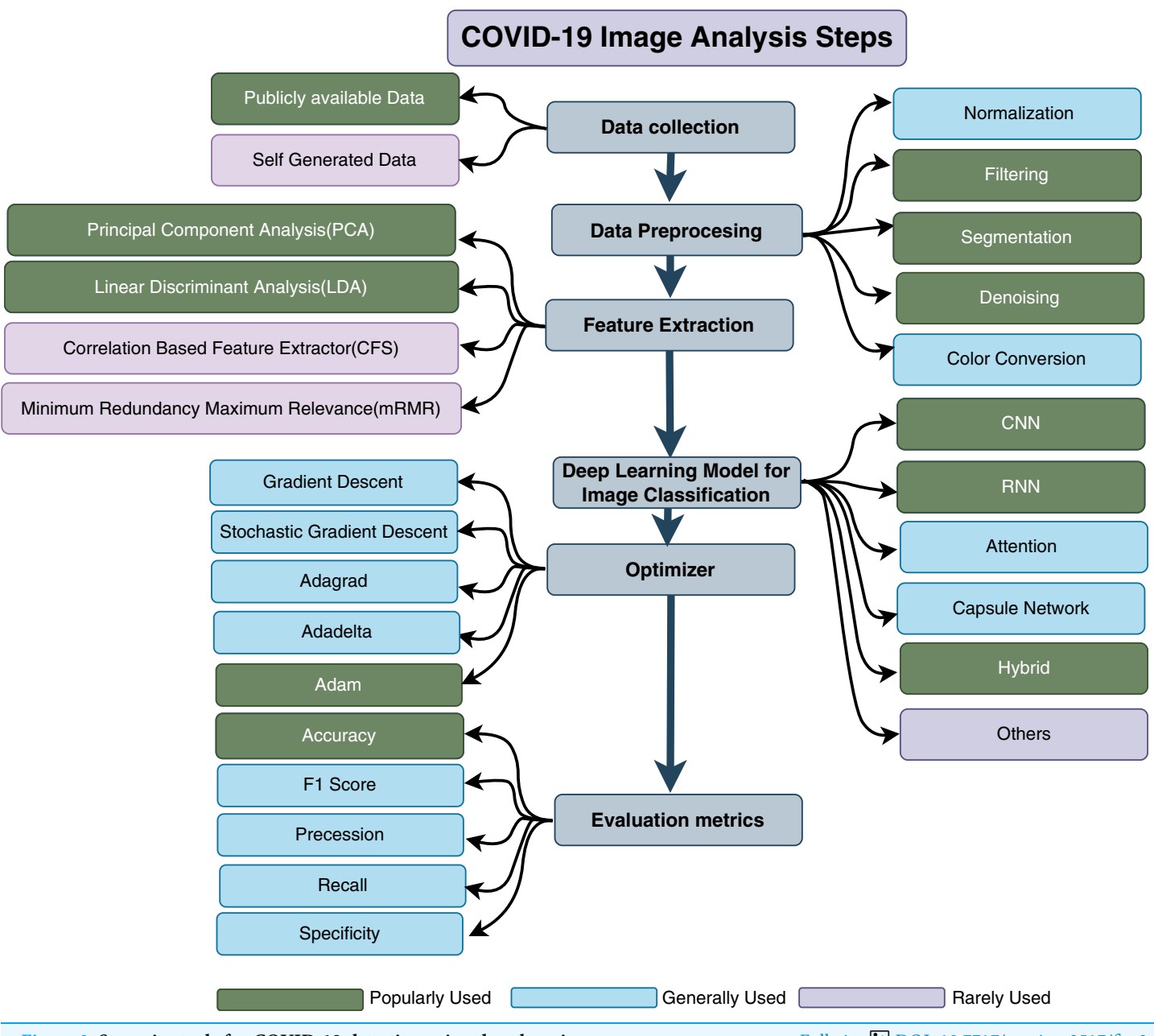

**Figure 3 Step wise tools for COVID-19 detection using deep learning.**

collaboration with other partners, gathering a compendium of research articles centred around SARS-CoV-2 studies (*Wang et al., 2020b*). A significant contribution by Berkeley Lab researchers involved the development of an online search interface catering to a comprehensive compilation of scientific journal data pertaining to SARS-CoV-2. This collection encompasses diverse scholarly datasets, including the works of *Wang et al. (2020b)*, Lite COVID, and the Elsevier Novel Coronavirus Resource Center.

Addressing the pandemic on a broader scale, the National Pandemic Initiative (NPI) amalgamates governmental efforts aimed at combatting SARS-CoV-2. To comprehensively

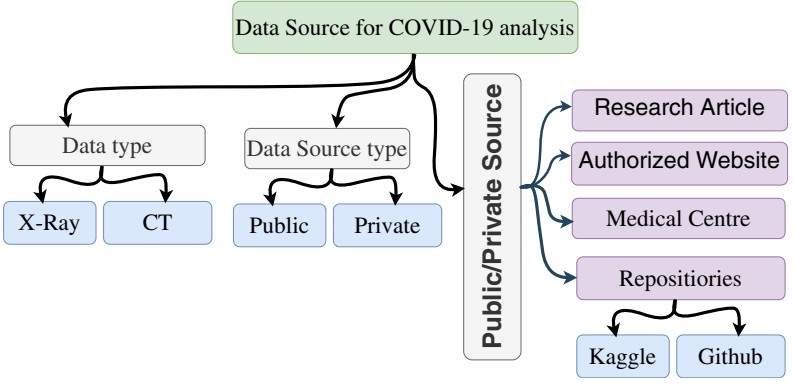

**Figure 4  Data source for SARS-CoV-2 analysis.**

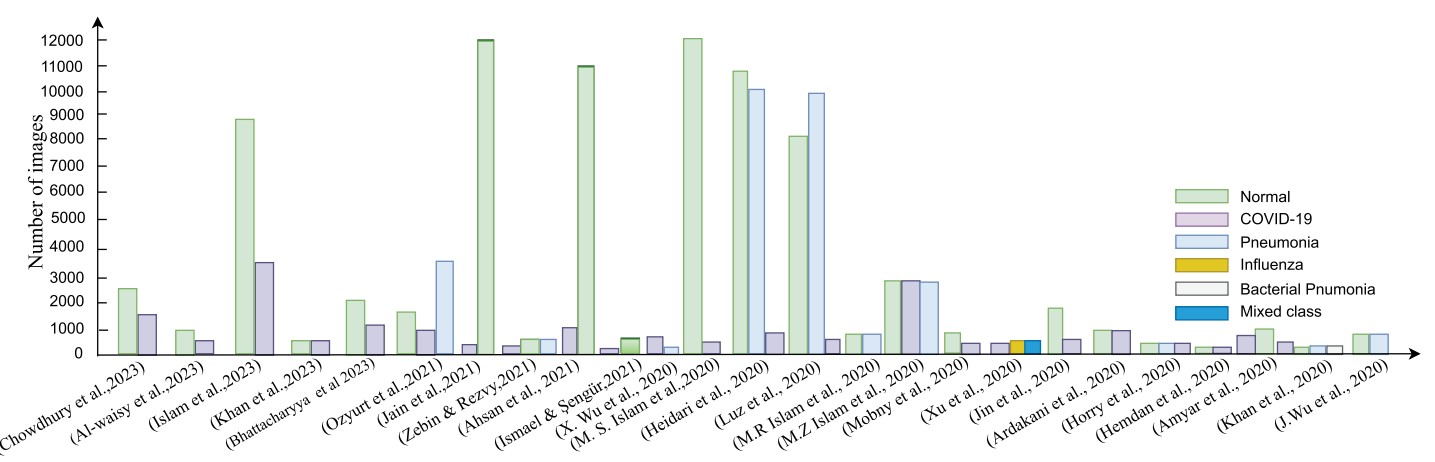

**Figure 5  The amount of image data used in each model for SARS-CoV-2 analysis using images.**

**Table 2  Public data sources.**

| Type | Amount | Class | Source |
|---|---|---|---|
| X-ray | 5,863 | 1,341 SARS-CoV-2 positive, 3,875 Viral Pneumonia | https://www.kaggle.com/datasets/paultimothymooney/chest-xray-pneumonia |
| X-ray | 454 | 137 SARS-CoV-2 and 317 Viral Pneumonia and Normal | https://www.kaggle.com/datasets/pranavraikokte/covid19-image-dataset |
| X-ray | 6,216 | 1,341, 4,875 | https://www.kaggle.com/datasets/paultimothymooney/chest-xray-pneumonia |
| X-ray | 325 | SARS-CoV-2, Non SARS-CoV-2 | https://github.com/ieee8023/covid-chestxray-dataset |
| CT, Xray | 1,124 | 403 SARS-CoV-2, 721 Normal | https://github.com/M4rc-C/OPENCovidGAN |
| Xray | N/A | SARS-CoV-2, Normal | https://www.kaggle.com/competitions/rsna-pneumonia-detection-challenge/data |
| Xray | N/A | SARS-CoV-2, Normal | https://github.com/agchung/Actualmed-COVID-chestxray-dataset |
| X-ray | 232 | 120 COVID, 112 Normal X-ray | https://twitter.com/chestimaging/ (requires Twitter login) |

(Continued)

| Type | Amount | Class | Source |
|---|---|---|---|
| X-ray (*Wang, Lin & Wong, 2020*) | 30,482 | 16,490 COVID, 13,992 Normal X-ray | https://www.kaggle.com/datasets/andyczhao/covidx-cxr2 |
| X-ray (*Wang, Lin & Wong, 2020*) | 100 | 40 COVID, 60 Normal X-ray | https://gitee.com/junma11/COVID-19-CT-Seg-Benchmark |
| CT scan | 746 | SARS-CoV-2: 349, non-SARS-CoV-2: 397 | https://github.com/desaisrkr/https-github.com-UCSD-AI4H-COVID-CT |

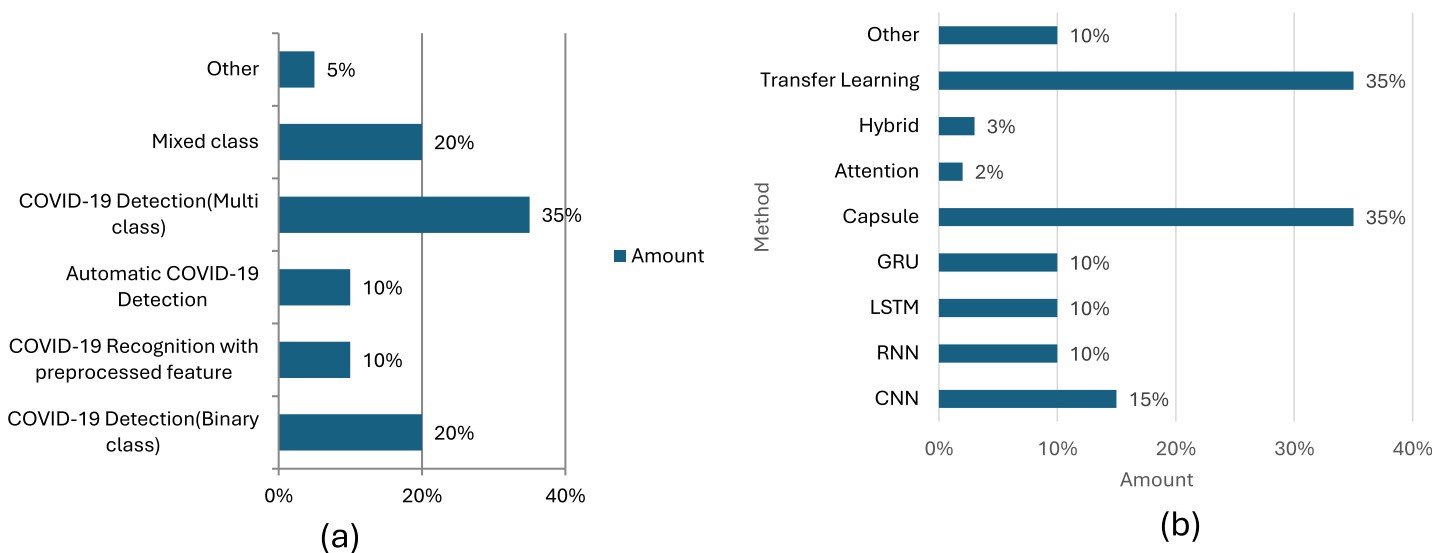

**Figure 6** (A) SARS-CoV-2 analysis amount based on category and (B) application-based amount.

study and interpret SARS-CoV-2 transmissions, substantial NPI datasets are requisite. Furthermore, investigations into the impact of NPI measures on SARS-CoV-2 incidents necessitate substantial volumes of data encompassing infections, fatalities, and related metrics.

In the quest for insights into SARS-CoV-2 transmission and the influence of NPI measures, a robust source of data comes in the form of the Stringency Index (*Hale et al., 2020*). Meticulously curated by a collective of professors and students from Oxford University, this index synthesizes openly accessible global data. Throughout the SARS-CoV-2 era, a significant portion of pertinent image data has been documented within a study journal (*Shuja et al., 2021*).

## Data pre-processiong

Data preprocessing is an essential step prior to its practical application. The goal of preprocessing is to refine raw data into a more polished dataset. Before applying any analytical method, an initial preprocessing of the dataset is carried out to identify and rectify missing values, inconsistent data, and other irregularities. This preliminary data

**Table 3 Data pre-processing tools for SARS-CoV-2 analysis.**

| Pre-processing | Purpose in SARS-CoV-2 detection |
|---|---|
| Color conversion | Color conversion (*Ozaltin, Yeniay & Subasi, 2023*), Pixel brightness transformations (*Mirza, Siddiq & Khan, 2023*), Brightness corrections (*Ebenezer et al., 2022*) |
| Normalization | Standardization of SARS-CoV-2 image (*Saiz & Barandiaran, 2020*), Geometric Transformations, Mean Normalization improves CNN accuracy in SARS-CoV-2 detection (*Yaman, Karakaya & Erol, 2022*), Normalization of SARS-CoV-2 CT image for feature extraction (*Pratiwi et al., 2021*), SARS-CoV-2image histogram equalization by CLAHE (*Al-Waisy et al., 2021*), Histogram Equalization increases performance of learning of CNN in SARS-CoV-2 Detection (*Maity, Nair & Chandra, 2020*). Image Restoration, Standardization: Standardization of SARS-CoV-2 image improves model learning to get better performance (*Saiz & Barandiaran, 2020*). A learning pipeline used for the standardization of SARS-CoV-2 chest X-ray images for deep learning method (*Wang et al., 2021a*) and Z-Component analysis. |
| Filtering | Smoothing (*Karthik, Menaka & Hariharan, 2021*), Edge Detection (*Purohit et al., 2022*), Sharpening (*Purohit et al., 2022*) Filtering CT (*Deshpande & Schuller, 2020*; *Subramanian et al., 2022*) and X ray image (*Han, Yang & Lee, 2010*; *Shorten, Khoshgoftaar & Furht, 2021*), Reshaping (*Fouladi et al., 2021*). |
| Segmentation | Segmenting SARS-CoV-2 image (*Yan et al., 2020*; *Wang et al., 2020a*), Fourier transform of SARS-CoV-2 image in deep learning (*Wang, Zhang & Zhang, 2021*). To make the more clear and formatted image as machine can understand SARS-CoV-2 detection with fine pre-processed image using CT and xray like Edge-based Segmentation (*Fan et al., 2020a*) Threshold Segmentation (*Voulodimos et al., 2021*), Region-based Segmentation (*Voulodimos et al., 2021*), Clustering-based Segmentation (*Tello-Mijares & Woo, 2021*) and Watershed-based Segmentation (*Ahsan et al., 2021*). |

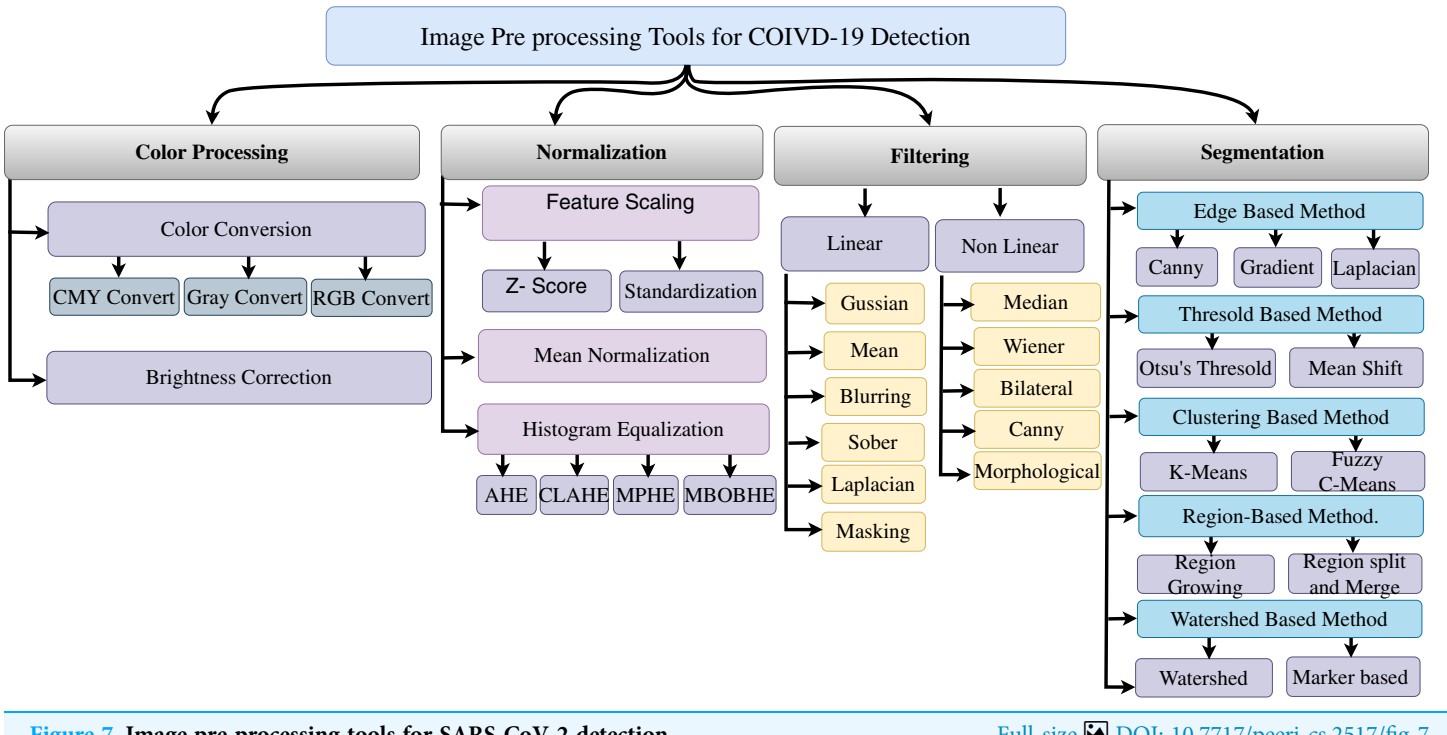

**Figure 7 Image pre-processing tools for SARS-CoV-2 detection.**

preprocessing stage plays a crucial role in the data mining process, involving data transformations or exclusions to ensure improved performance and reliability.

In Table 3 and Fig. 7, we provide insights into the nature of preprocessing tools utilized in SARS-CoV-2 detection from image data. These visual representations shed light on the

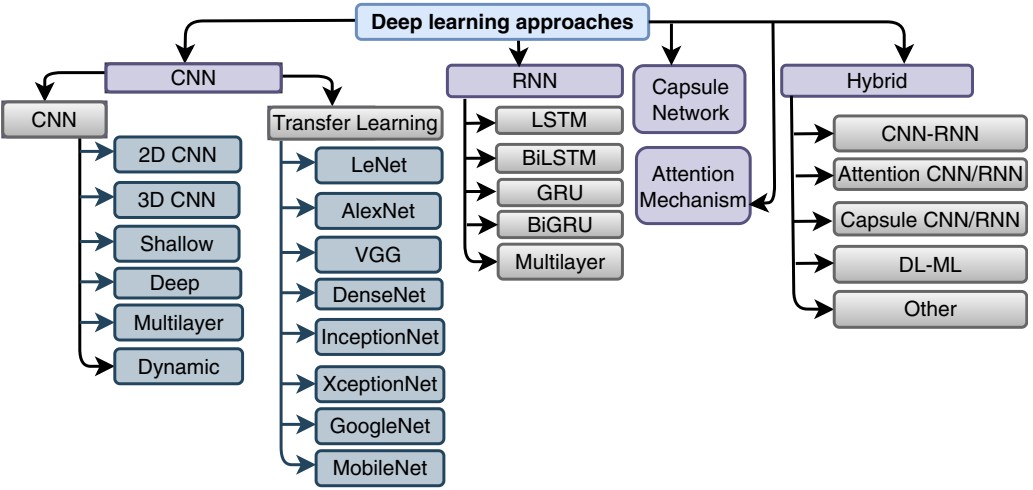

**Figure 8 Taxonomy of SARS-CoV-2 analysis using deep learning method.**

specific preprocessing techniques employed to enhance data quality and suitability for subsequent analyses.

## Types of deep learning approaches

The realm of SARS-CoV-2-related technical analysis is characterized by a multitude of methods and approaches. Within this section, we delve into the comprehensive taxonomy of SARS-CoV-2 analysis methodologies, which is vividly depicted in Fig. 8. This visual representation provides a holistic outlook on the spectrum of deep learning approaches harnessed for SARS-CoV-2 detection, analysis, and classification. The analytical pathways for SARS-CoV-2 encompass conventional techniques (statistical or manual), traditional machine learning paradigms (such as SVM (*Kohmer et al., 2020*; *Ismael & Şengür, 2021*) and random forest (*Wu et al., 2020b*)), as well as deep learning methods.

Central to our focus is the deep learning-based approach to SARS-CoV-2 analysis. This method is further categorized into supervised, unsupervised, and semi-supervised modalities, reflecting a comprehensive framework for harnessing the potential of deep learning more in combating the challenges posed by SARS-CoV-2.

Recently, popularly used methods are convolutional neural network (CNN) (*Akl et al., 2023*; *Kaya & Gürsoy, 2023*; *Kuzinkovas & Clement, 2023*; *Choudhary et al., 2023*; *Khan et al., 2023*; *Sohrabi et al., 2020*; *Chakraborty, Murali & Mitra, 2022*; *Bhattacharyya et al., 2022*; *Islam et al., 2020*), recurrent neural network (RNN) (*Rayan & Alaerjan, 2023*; *AlMohimeed et al., 2023*; *Raamkumar, Tan & Wee, 2020*) and Hybrid (*Islam et al., 2020*; *Shah et al., 2021*; *Islam, Islam & Asraf, 2020*). CNN-based SARS-CoV-2 diagnosis and prediction is the most popular. CNN method may be hybridized with attention (*Ullah et al., 2023*; *Ouyang et al., 2020*; *Yang et al., 2023*; *Wen et al., 2023*; *Christina Magneta, Sundar & Thanabal, 2023*), capsule, shallow, deep (*Lella & Pja, 2022*; *Wu et al., 2020a*), dynamic, deep CNN (*Ardakani et al., 2020*; *Ozyurt, Tuncer & Subasi, 2021*) and multilayer, hybrid CNN (*Laguarta, Hueto & Subirana, 2020*; *Jain et al., 2021*). In the cased of RNN

(*Imran et al., 2020*; *Hassan, Shahin & Alsabek, 2020*; *Imran et al., 2020*), it has two evolutions named long short-term memory (LSTM) (*Zhang et al., 2020*) and gated recurrent unit (GRU) (*Raamkumar, Tan & Wee, 2020*). Both LSTM (*Islam et al., 2020*) and GRU (*Paka et al., 2021*) may be applied in bidirectional (*Passricha & Aggarwal, 2020*). Capsule network with RNN also gives good outcomes (capsule network) (*Yuan et al., 2023*; *Sharma et al., 2023*; *Wen et al., 2023*; *Zhao et al., 2023*; *Islam et al., 2020*; *Prabha & Rathipriya, 2020*; *Malla & Alphonse, 2021*; *Mobiny et al., 2020*; *Afshar et al., 2020*). Generalized adversarial network (GAN) (*Bar-El et al., 2021*; *Fan et al., 2020b*; *Elzeki et al., 2021*; *Goel et al., 2021*; *Quilodrán-Casas et al., 2022*; *Zhu et al., 2020*; *Fan et al., 2020b*) model works good than other method. Attention mechanism (*Sitaula & Hossain, 2021*; *Zhang et al., 2021*; *Paka et al., 2021*; *Zhang et al., 2020*; *Wang et al., 2020c*; *Pinkas et al., 2020*) sometimes obtained very good performance in SARS-CoV-2 detection. Graph conventional network (GCN) (*Yu et al., 2021*; *Kapoor et al., 2020*; *Che et al., 2021*; *Wang et al., 2021b*; *Laguarta & Subirana, 2020*; *Liang et al., 2021*) also applied on SARS-CoV-2 detection. On the other hand, ResNet (*Al-Waisy et al., 2023*; *Islam et al., 2023a*), Light CNN (*Khan et al., 2023*), dense and convolution (*Ulukaya et al., 2023*), Transfer BERT (*Contreras Hernández et al., 2023*; *Qorib et al., 2023*), RNN, multilayer perception (MLP), CNN, (*Paul, Saha & Singh, 2023*) *etc* gives good outcome in SARS-CoV-2 detection.

Deep learning algorithms can help solve complex problems by analysing fundamental depictions. Several layers are utilised progressively to learn exact interpretations as well as the property of annotated training data. Deep learning (*Ardakani et al., 2020*) algorithms applied in health facilities (*Kong et al., 2020*), smart healthcare (*Esteva et al., 2019*), drug development (*Chen et al., 2018*), medical image recognition (*Kim et al., 2020*), and so on. It is also widely used in the automatic diagnosis of SARS-CoV-2 victims or patients. Convolutional neural network (*Ardakani et al., 2020*) is applied in different transfer learning (*Ahsan et al., 2021*; *Wu et al., 2020a*). Transfer learning models are used widely those are known as such as AlexNet (*Ardakani et al., 2020*), GoogleNet (*Ardakani et al., 2020*), SqueezeNet (*Ardakani et al., 2020*), different versions VGG (*Zebin & Rezvy, 2021*; *Hemdan, Shouman & Karar, 2020*), diverse kinds of ResNet *Wu et al. (2020b)* (*Wu et al., 2020b*; *Xu et al., 2020*), Xception (*Ardakani et al., 2020*; *Hemdan, Shouman & Karar, 2020*), inception (*Jain et al., 2021*; *Horry et al., 2020*; *Hemdan, Shouman & Karar, 2020*), diverse types of MobileNet (*Ardakani et al., 2020*), DenseNet (*Hemdan, Shouman & Karar, 2020*), encoder-decoder (*Amyar et al., 2020*), capsule network (*Afshar et al., 2020*) *etc*.

## Performance metrics used in COVID-19 detection

This section encompasses the key performance measures pertinent to SARS-CoV-2 analysis. Performance metrics serve as indicators of the efficacy of a given strategy. The presence of various variables and highly imbalanced datasets, stemming from diverse factors, underscores the significance of accurately gauging the annotations authenticity across the entire dataset. Consequently, these distinctive variables may not be directly comparable to conventional benchmarks.

The efficacy study seeks to demonstrate the operational capability of a system. Within the realm of SARS-CoV-2 analysis, several performance indicators are employed, including precision, recall, F-1 score, specificity, and sensitivity. These indicators facilitate the quantification of outcomes based on comparisons between anticipated and observed results. In Fig. 3, a subset of factors utilized to assess the landscape of SARS-CoV-2 research is illustrated. This depiction highlights the ongoing efforts in deep learning-based SARS-CoV-2 research, emphasizing the multifaceted evaluation process encompassing diverse aspects of performance.

Performance metrics are part of every machine learning process evaluation. Actually, it tells you if the model is developing and assigns a number to it. Deep learning techniques are indeed examples of machine learning models. Several performance metrics are used to judge the effectiveness of a deep learning model. The most common evaluation measures are listed here. P denotes positive, T indicates true, F indicates false, as well as N indicates negative in the calculation. The projected value $y_i$ in the loss function is y. The equations are used to calculate their predictions (Eqs. (1) to (5)).

$$Accuracy = \frac{(TP + TN)}{TP + TN + FP + FN} \tag{1}$$

$$Precision = \frac{(TP)}{TP + FP} \tag{2}$$

$$Recall = \frac{(TP)}{TP + FN} \tag{3}$$

$$F1 = 2 * \frac{(Precession * Recall)}{Precession + Recall} \tag{4}$$

$$Loss(y) = y_i log(\bar{y}_i) + (1 - y_i) + log(1 - \bar{y}_i) \tag{5}$$

## DEEP LEARNING METHODS

Within this section, we will delve into the latest advancements in deep learning methods specifically applied to the analysis and detection of SARS-CoV-2. Our categorization organizes deep learning methods into three distinct categories, focusing on models that leverage image data. Comprehensive insights into each model are provided, encompassing visual representations, mathematical equations, in-depth analyses, and intricate detection mechanisms. Basic deep-learning models like CNN and RNN are foundational tools for image analysis. Additionally, we explore more advanced deep learning approaches such as transfer learning, attention-based mechanisms, capsule networks, and the fusion of CNN and RNN architectures. In particular, a thorough exploration of the CNN model is presented, offering a comprehensive understanding of its internal workings.

Before delving into intricate details, it is essential to establish the fundamental steps and prerequisites of image-based SARS-CoV-2 analysis. This foundational understanding sets the stage for a comprehensive exploration of the various deep learning models employed in this critical domain.

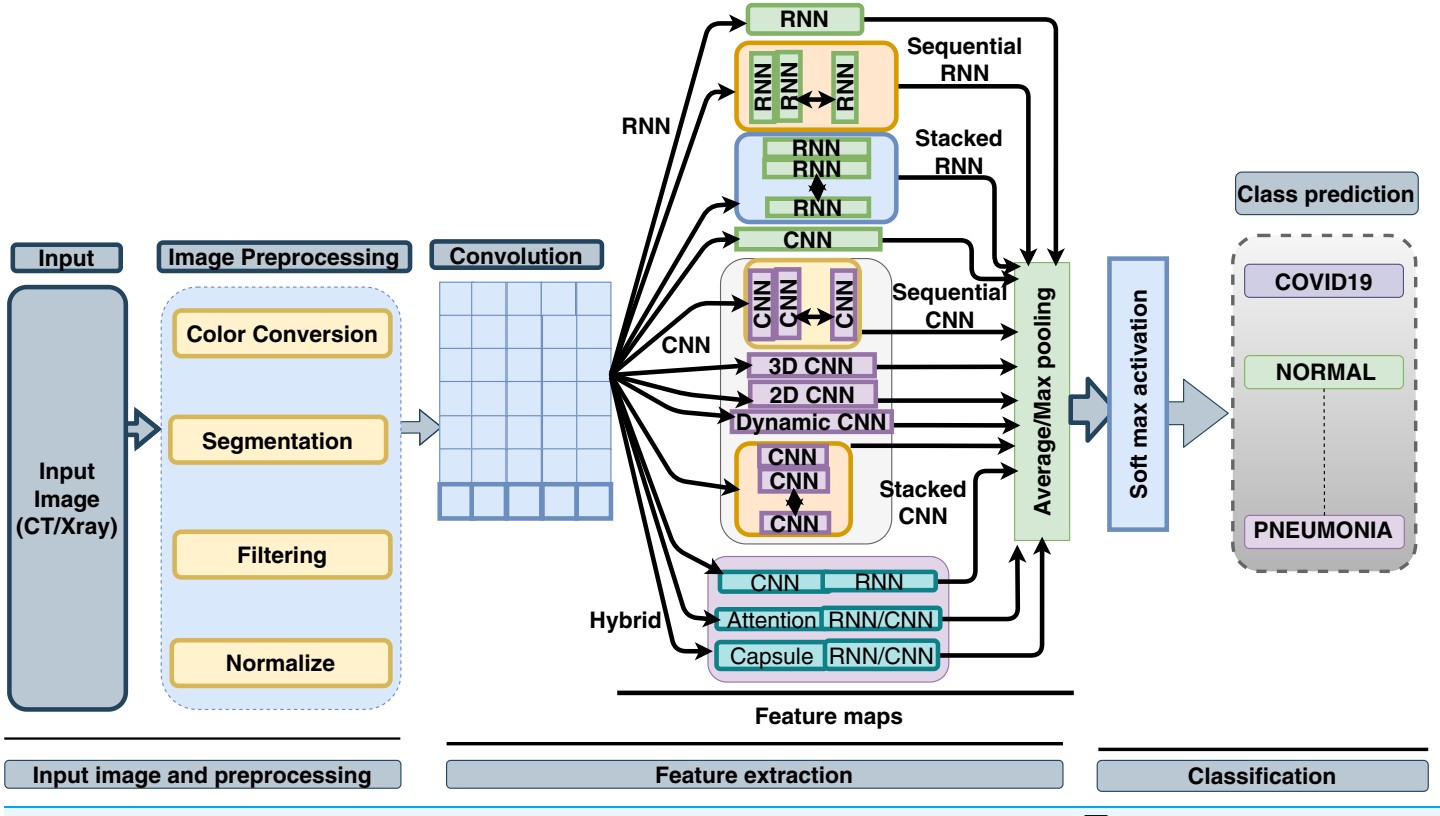

**Figure 9** Overall diagram of all deep learning methods to analyse SARS-CoV-2 from image.

## Basic operating step in deep learning based COVID-19 detection

Figure 9 provides an overview of the fundamental pipeline activities encompassing SARS-CoV-2 diagnosis and detection systems driven by deep neural networks. Image-based SARS-CoV-2 analysis stands as a prominent and accurate avenue of SARS-CoV-2 research. Within Fig. 9, we outline several key steps integral to image-based SARS-CoV-2 detection. Commencing with data collection from diverse sources, including open, private, and crowd-sourced origins, the participation of medical community patients is emphasized from the outset. This data, available in various formats, often employs imaging modalities such as CT scans and X-ray imaging for SARS-CoV-2 diagnosis.

A novel pre-processing methodology enhances predictive capabilities in SARS-CoV-2 analysis based on labelled image data. This involves various sub-processes like normalization, standardization, and Zero Component Analysis, all contributing to data refinement. Specifically tailored to SARS-CoV-2 imaging data comprising CT and X-ray images, the data pre-processing stage serves to eliminate undesired or extraneous elements from the images. Subsequently, pre-processed data undergoes feature analysis, encompassing feature extraction, selection, and matching. Deep learning algorithms engage in data analysis to classify SARS-CoV-2 cases following the selection of pertinent

features. Within the classification phase, sub-steps encompass data partitioning, model or method selection, parameter tuning, and categorization. Data is divided into three distinct sets: training, validation, and testing, often utilizing the widely employed cross-validation technique. A model is constructed using training data, subsequently validated with the validation set, and ultimately tested using the dedicated test data. Central to the realm of computer learning-based SARS-CoV-2 diagnosis, the feature extraction and classification steps, facilitated by deep learning techniques, play a pivotal role. In this phase, deep learning automatically extracts distinctive features through iterative processes, followed by classification based on target or output classes (non-SARS-CoV-2 or SARS-CoV-2). The evaluation of the developed system's performance encompasses metrics such as accuracy, recall, sensitivity, specificity, precision, and F1-score.

Figure 9 encapsulates the comprehensive framework of all deep learning algorithms employed for SARS-CoV-2 image analysis. At the figure's inception, diverse forms of data are input, which are subsequently pre-processed based on their respective categories. Following pre-processing, the deep model is engaged. Deep learning models fall into several categories, including RNN, CNN, Attention, Capsule, Hybrid, and many others. Sequential implementation denotes the use of multiple levels of CNN or RNN in a consecutive sequence, which can also be layered for more intricate analysis.

## CNN method

CNN is a simple image classification algorithm that is widely used. CNN is also used to classify images. *LeCun et al. (1998)* proposed the CNN structure for the first time (*LeCun et al., 1998*). Small data chunks are converted into conventional high-level vectors *via* a convolution stratum. The fundamental setup and corresponding CNN layout layers are shown in Fig. 10A. We also offer a brief overview of the recently built deep learning-based CNN model (*Akl et al., 2023*; *Kaya & Gürsoy, 2023*; *Kuzinkovas & Clement, 2023*; *Choudhary et al., 2023*; *Khan et al., 2023*) for SARS-CoV-2 analysis in the table. In classification tasks, CNN-based transfer learning methods are very common and practical. CNN-based approaches have been widely used in SARS-CoV-2 identification and classification from images in recent years. *Jain et al. (2021)* have developed a new method for detecting SARS-CoV-2 using a CNN classifier. This method produces a 97 percent accuracy rate on average, however, it only works with limited data and has overfitting.

Here we present the recent CNN methods of SARS-CoV-2 analysis from image (*Akl et al., 2023*; *Kaya & Gürsoy, 2023*; *Kuzinkovas & Clement, 2023*; *Choudhary et al., 2023*; *Khan et al., 2023*; *Islam et al., 2020*; *Sohrabi et al., 2020*; *Luz et al., 2021*); Fig. 11 depicts the overall process of the SARS-CoV-2 study employing all CNN-based models derived from image data. In this diagram, data is preprocessed individually before being transmitted to a CNN-based transfer learning-based system. To indicate covid as well as non-covid class, the outcome of model execution. The actual structure of each CNN model (Transfer-based) is illustrated in this diagram.

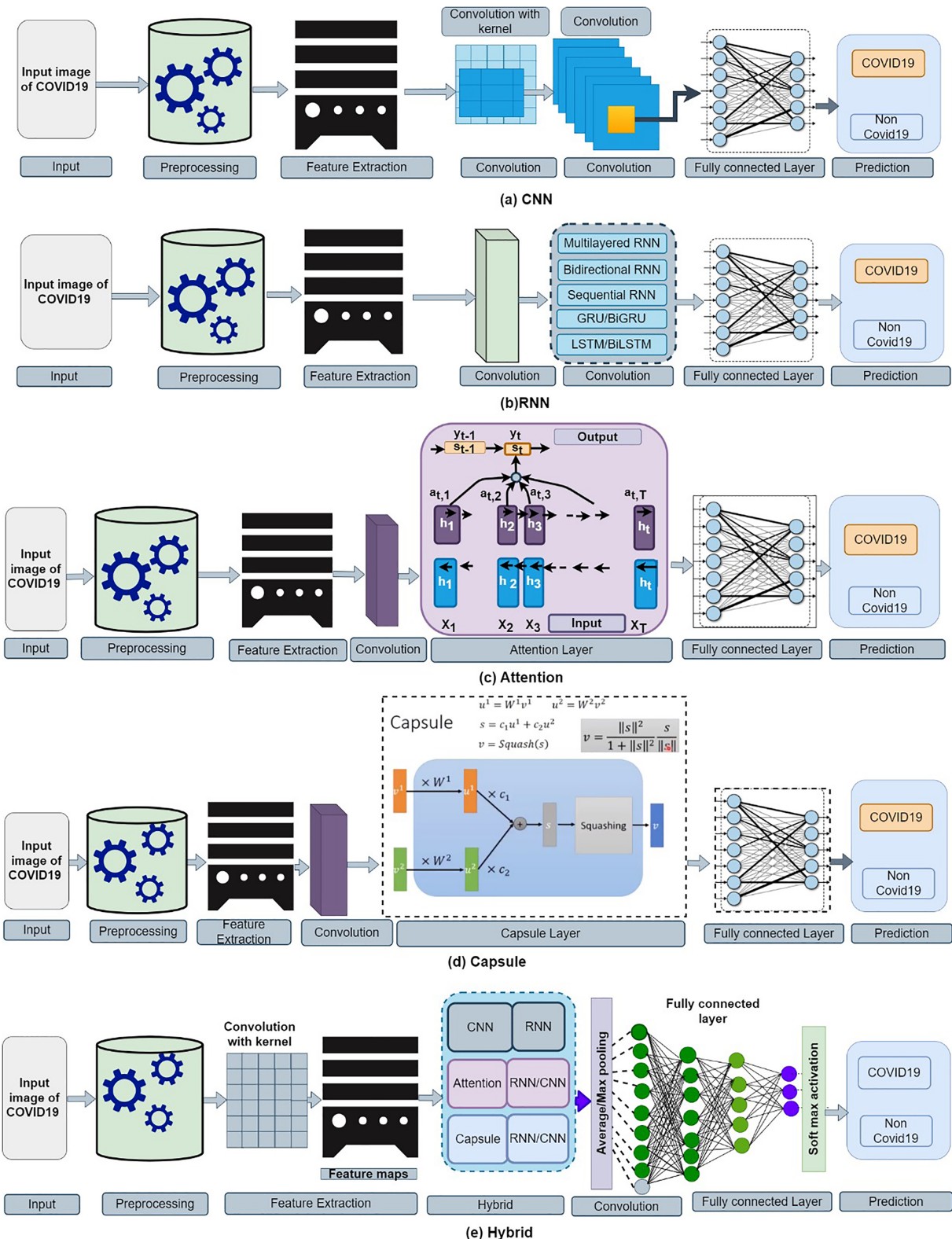

**Figure 10 Operational figure: (A) CNN (B) RNN (C) attention (D) capsule and (E) hybrid-based model for SARS-CoV-2 analysis from the image.**

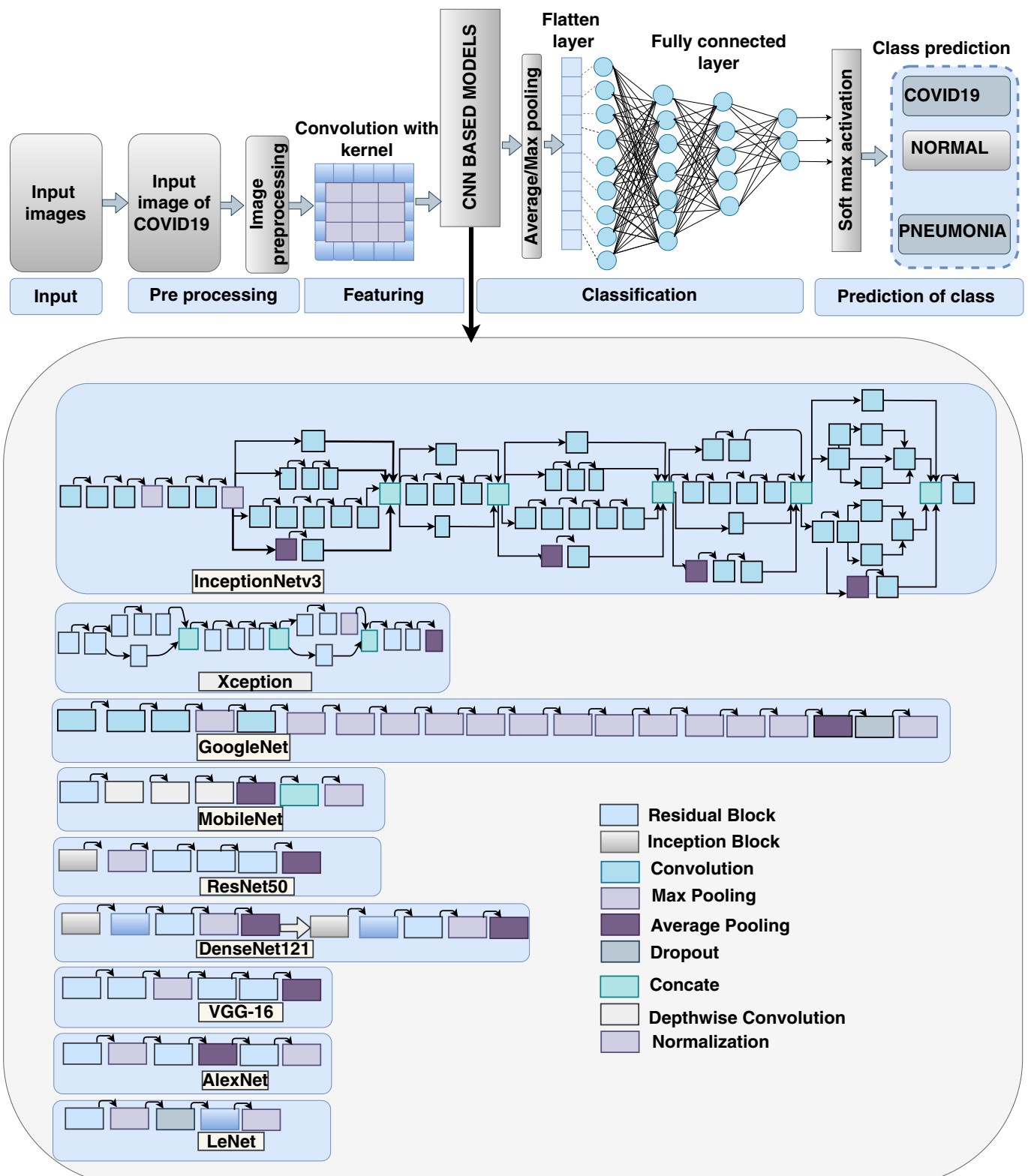

**Figure 11 Operational diagram of transfer learning method: InceptionNetV3, XceptionNet, GoogleNet, MobileNet, ResNet50, DenseNet121, VGG-16, AlexNet and LeNet.**

### Transfer learning method

Transfer learning is indeed a supervised learning method that allows us to reuse a here-for-learning algorithm as the foundation for a new model on a different challenge. Simply put, a model acquired on one task is repeated on a comparable assignment as an enhancement, allowing for faster modeling development on the second activity. The overall process diagram to present the working flow of transfer learning is given in Fig. 11.

A LeNet-based CNN classifier (*Islam & Matin, 2020*) was employed to classify tiny SARS-CoV-2 CT image data that obtained 56 percent accuracy. However, this LeNet is not suitable for huge datasets. *Zebin & Rezvy (2021)* used VGG16, ResNet50, and EfficientNetB (*Zebin & Rezvy, 2021*) models to detect SARS-CoV-2 in 822 X-ray pictures and achieved accuracy of 90 percent, 94.3 percent, and 96.8 percent, correspondingly. However, this strategy is difficult to use. Another method using ResNet and SVM achieved a 94.7 percent accuracy rate, although it only works with a limited dataset. With a 76 percent accuracy, 81.1 percent sensitivity, and 61.5 percent specificity, and an area under the curve (AUC) of 81.9 percent, a multifunctional based deep learning algorithm with ResNet50 was used. To analyse medical evidence for screening SARS-CoV-2. SARS-CoV-2 detection with 94.98 percent accuracy using the ResNet52 model and thorough statistical category analysis. However, instead of lesion segmentation, our method removes only the attentional region. Another study that used practically all CNN-based classifiers demonstrated their effectiveness and recommended the ResNet model as the best model (*Ardakani et al., 2020*). Another new technique to assess SARS-CoV-2 is Coronet by *Sohrabi et al. (2020)* with an accuracy of 89.5 percent, the sensitivity of 100 percent, precision of 97 percent, and F1-score of 98 percent.

- LeNet: In 1989, *LeCun et al. (1989)* developed the LeNet CNN structure. A recent method for analysing SARS-CoV-2 using the LeNet model yielded an accuracy of 56.06 percent (*Islam & Matin, 2020*). Figure 11 depicts the fundamental structure of LeNet for analysing SARS-CoV-2 from the image.
- AlexNet: AlexNet (*Krizhevsky, Sutskever & Hinton, 2012*) contained eight layers, with the first five being convolution layers, followed by max-pooling layers in some settings, and the last three being completely linked layers, as seen in Fig. 11. Which used the non-saturating ReLU activation function, which had better training accuracy than tanh and sigmoid. AlexNetcite19 for image-based SARS-CoV-2 analysis. Figure 11 shows the fundamental structure of AlexNet (*Ardakani et al., 2020*), which is used to analyse SARS-CoV-2 from images.
- VGG: The Visual Geometry Group Network (VGG) (*Simonyan & Zisserman, 2014*) model does a good job at detecting SARS-CoV-2. ResNet (*Ardakani et al., 2020*) has observed recently SARS-CoV-2. Figure 11 shows the fundamental structure of VGG-16, which may be used to analyse SARS-CoV-2 from a picture.
- DenseNet: The "Dense Convolutional Net" (*Huang et al., 2017*) (DenseNet) is a relatively new concept in neural nets for visual object recognition. SARS-CoV-2 detection is accomplished with success using DenseNet (*Horry et al., 2020*). Figure 11

shows the fundamental structure of VGG-16, which may be used to analyse SARS-CoV-2 from a picture.

- InceptionNet: An Inception Module (*Szegedy et al., 2016*) in SARS-CoV-2 detection is accomplished with success utilizing Inception (*Horry et al., 2020*). Figure 11 shows the fundamental structure of VGG-16, which may be used to analyse SARS-CoV-2 from a picture.

- ResNet: ResNet (*Szegedy et al., 2016*) is comparable to VGGNet, but it is eight times more profound (*Too et al., 2019*). SARS-CoV-2 was successfully evaluated with around 95 percent correctness by ResNet (*Ardakani et al., 2020*). Figure 11 depicts the fundamental structure of ResNet for analysing SARS-CoV-2 from an image.

- Inception-ResNet-V2: Inception V3 (*Szegedy et al., 2017*) in SARS-CoV-2 detection has achieved favorable performance using the Inception as well as ReNet model together (*Horry et al., 2020*). Figure 11 depicts Inception-ResNet basic model.

- XceptionNet: The Xception framework (*Chollet, 2017*) in SARS-CoV-2 detection with Xception yields a satisfactory result (*Horry et al., 2020*). Figure 11 shows the fundamental structure of XceptionNet, which is used to analyse SARS-CoV-2 from images.

- GoogleNet: GoogleNet utilises Inception modules, which facilitate the network to choose from a wide range of convolutional filter sizes within every block. Figure 11 shows the GoogleNet convolutional neural network, which is based on the Inception architecture and is a type of convolutional neural network. GoogleNet is now detecting SARS-CoV-2 successfully. Figure 8 depicts the fundamental structure of GoogleNet for analysing SARS-CoV-2 from an image. Detection of SARS-CoV-2 from an X-ray image by utilising GoogLeNet-COD got 87.5% accuracy (*Yu et al., 2020*).

- MobileNet: MobileNet is a reduced design that uses depth-wise separable CNN layers to build basic deep neural networks (*Howard et al., 2017*) in many standard deeper learning systems. Figure 11 shows the fundamental operating sequence of MobileNet to analyse SARS-CoV-2 from the image in a new application of MobileNet to identify SARS-CoV-2 (*Ardakani et al., 2020*).

## RNN method

In deep learning, RNN can handle long-ranged image features. Figure 10B depicts the basic structure of an RNN-based SARS-CoV-2 study. We also offer a brief overview of the recently built deep learning-based RNN model (*Rayan & Alaerjan, 2023*; *AlMohimeed et al., 2023*) for SARS-CoV-2 analysis in the table.

In SARS-CoV-2 case predictions utilising LSTM and RNN in GCC countries and India: Lethality and tests with several hidden layers, an LSTM model has a high degree of precision and a tiny prediction error (*Razia, Arokiaraj & Jaithunbi, 2021*). An RNN-based technique with CNN detected SARS-CoV-2 (*Islam, Islam & Asraf, 2020*) with 99.4 percent accuracy from 4,575 pictures (*Islam, Islam & Asraf, 2020*).

## Attention method

Attention-based deep learning models (*Bahdanau, Cho & Bengio, 2014*) are better at handling mot relevant features than some other approaches. SARS-CoV-2 was detected with 87.49 percent accuracy using an attention-based VGG-16 classifier from 8,571 chest X-ray images (*Sitaula & Hossain, 2021*). This method's data augmentation should be enhanced. SARS-CoV-2 CT images with higher precession of 94.6 percent (*Zhang et al., 2021*) were segmented using a dense based GAN as well as multi-layer attention model with a decent segmentation approach. The overall process diagram of attention-based SARS-CoV-2 detection is presented in Fig. 10C. We also offer a brief overview of the recently built deep learning-based attention model (*Ullah et al., 2023*; *Ouyang et al., 2020*; *Yang et al., 2023*; *Wen et al., 2023*; *Christina Magneta, Sundar & Thanabal, 2023*) for SARS-CoV-2 analysis in the table.

## Capsule network method

The capsule and its principal capsule layer give a high level of integration for collecting as well as maintaining a complex hierarchy parallel to the various kernel dimensions (*Mobiny et al., 2020*). Figure 10D shows how the output is transferred to the capsule's output layer. Overall process diagram of capsule network-based SARS-CoV-2 detection is presented in Fig. 10D. We also offer a brief overview of the recently built deep learning-based capsule model (*Yuan et al., 2023*; *Sharma et al., 2023*; *Wen et al., 2023*; *Zhao et al., 2023*; *Sejuti & Islam, 2023*; *Basha et al., 2023*; *Pustokhin et al., 2023*; *Deepak et al., 2023*) for SARS-CoV-2 analysis in the table.

A capsule network technique can be used in conjunction with RNN to improve performance. The capsule-based technique, which was recently applied in SARS-CoV-2 analysis, saves training time (*Afshar et al., 2020*). A capsule network might help you save time during training. SARS-CoV-2 was diagnosed with 95.7 percent accuracy from X-ray pictures using a capsule network from 13,800 images of COVIDx data (*Afshar et al., 2020*). A novel Radiologist SARS-CoV-2 CT scan image detection with full capsule systems with a performance of 87.6 percent accuracy (*Mobiny et al., 2020*). This system only works with 746 photos on a tiny domain (*Mobiny et al., 2020*).

## Hybrid method

A hybrid deep learning-based method is constructed employing CNN, RNN, as well as other techniques, as illustrated in Fig. 10E. We offer a brief overview of the recently built deep learning-based hybrid model (*Sejuti & Islam, 2023*; *Basha et al., 2023*; *Pustokhin et al., 2023*; *Deepak et al., 2023*) for SARS-CoV-2 analysis in the table. This section describes some of the most up-to-date image classifiers required to complete the clinic intent. This section demonstrates how to analyse SARS-CoV-2 utilizing images using a recently discovered deep learning algorithm.

SARS-CoV-2 detection from chest 424 X-ray pictures with a precision of 96.00 percent (*Shah et al., 2021*) that used a unique Deep GRU-CNN algorithm. A deep residual network-based type attention layer with bidirectional LSTM was created for the identification and diagnosis of SARS-CoV-2. RCAL-BiLSTM is a novel ResNet-based Class

Attention Layer with Bidirectional LSTM that obtains a SARS-CoV-2 diagnosis (*Pustokhin et al., 2020*) accuracy of 94.88 percent. SARS-CoV-2 (*Islam et al., 2020*) was detected using a hybrid deep learning approach with 40,000 pictures. This approach had a 99.4 percent accuracy rate. On 20,000 pictures, a deep learning-based model combining ANN to SARS-CoV-2 including feature extraction achieved 95.84 percent accuracy (*Ozyurt, Tuncer & Subasi, 2021*). SARS-CoV-2 was detected with 99.2 percent accuracy using a CNN-LSTM-based algorithm that classified 4,575 images (*Islam, Islam & Asraf, 2020*).

# RESULT ANALYSIS

This section is dedicated to the examination of comparison results. Structurally, it comprises two distinct components: firstly, a comprehensive analysis of comparative outcomes stemming from contemporary approaches as discussed in the review, and secondly, an in-depth analysis of results derived from our own manual execution of methods.

## Comparative analysis

This section encompasses a comprehensive analysis of the overall comparison results. Undoubtedly, this segment holds significant importance within the scope of this article. Our scrutiny has been meticulously directed towards the pivotal articles in the comparative analysis, aiming to illuminate their conclusions and underscore the existing gaps in research. Inclusive within this discussion are intricate details encompassing approaches, datasets, validation procedures, and other pertinent facets.

The crux of the comparative results is encapsulated within Tables 4–8. In these tables, abbreviations are employed for succinct representation: CV signifies cross-validation, AC corresponds to accuracy, S stands for sensitivity, P denotes precision, R represents recall, T denotes training, TE represents testing, and V signifies validation. Moreover, the letters C and SP denote the number of classes and Specificity, respectively. These tables collectively provide a comprehensive visualization of the main comparative outcomes, enriching our understanding of the performance and characteristics of various methods.

## Experimental analysis for model recommendation

We conducted a manual execution of 12 distinct models, meticulously tracking their individual accuracy and loss function throughout both the training and validation phases. This rigorous process was undertaken to thoroughly assess and validate the performance of newly developed models. To facilitate this evaluation, we employed the SARS-CoV-2 database, which primarily comprises X-ray images, serving as a robust foundation for comprehensively scrutinizing each model's capabilities. The SARS-CoV-2 data used for our experimental analysis can be found at Kaggle site (https://www.kaggle.com/tawsifurrahman/COVID19-radiography-database) (*Tawsif, 2022*). We use 4,000 X-ray images and 1,000 photos for four image classes: normal, viral pneumonia, lung opacity, as well as SARS-CoV-2. Table 9 displays the training and validation accuracy as well as loss of twelve distinct deep learning models over 10 epochs.

**Table 4 CNN and transfer learning method for SARS-CoV-2 analysis.**

| Author | Data type | Method | Data size | C | V | T and V size | Result | Key contribution and Findings | Research gaps |
|---|---|---|---|---|---|---|---|---|---|
| Akl et al. (2023) | X-ray image | Hybrid CNN | 2,482 | 2 | CV | T 80%: V 20% | AC: 99.39% | Scale-Invariant Feature Transform (SIFT) used for feature extraction and Hybrid deep learning boost detection performance | Real time detection is not proposed |
| Kaya & Gürsoy (2023) | X-ray image | MobileNet | 9,457 | 3 | 5k-CV | T 80%: V 20% | AC: 95.62%, 96.10%, 97.61% | Scale-Invariant Feature Transform (SIFT) used for feature extraction and Hybrid deep learning boost detection performance | Real time detection is not proposed |
| Kuzinkovas & Clement (2023) | X-ray image | Ensemble CNN | 33,920 | 2 | CV | T 80%: V 20% | AC: 98.34% | ResNet and VGGG pre-trained models are used for the prediction | Generalizability is weak |
| Choudhary et al. (2023) | X-ray image | ResNet | 2,482 | 2 | CV | T 80%: V 20% | AC: 95.47% | Incorporation of a ResNet34 high resolution network model utilized in SARS-CoV-2 detection | Did not show real-time image prediction |
| Khan et al. (2023) | X-ray image | Light CNN | 380 | 2 | 5K-CV | T 80%: V 20% | AC: 98.8% | Novel light CNN model using watershed region-growing segmentation for Chest X-rays image | Comparative Performance low when working with multiclass data. |
| Chakraborty, Murali & Mitra (2022) | X-ray | ResNet | 10,040 | 3 | CV | T 80%: V 20% | AC: 96.43% S: 93.68% | Detect SARS-CoV-2 efficiently | Lack of model validation Data set is small |
| Bhattacharyya et al. (2022) | X-ray | VGG-19, CGAN | 3,750 | 3 | CV | T 80%: V 20% | AC: 96.6% | Automatic SARS-CoV-2 detection with CNN model | Overfitting and under fitting |
| Aggarwal et al. (2022) | X-ray | Transfer Learning | 959 | 4 | CV | T 80%: V 20% | AC: 97% | Efficiently used transfer learning for SARS-CoV-2 analysis | Learning on small data, Performance low. |
| Islam et al. (2022) | X-ray | DenseNet | 231 | 3 | CV | T 80%: V 20% | AC: 96.49% | Properly utilized CNN in COVID recognition. | Overfitting in performance and used very Small data |
| Jain et al. (2021) | X-ray | Inception V3/ Xception Net/ResNet | 6,432 | 3 | CV | T 80%: V 20% | P: 99% R: 92% F1: 95% | Identify SARS-CoV-2 by CNN properly | Under fitting problem Small data |
| Zebin & Rezvy (2021) | X-ray | VGG16, ResNet50, Efficient Net | 822 | 3 | 5 Fold | T 80%: V 20% | AC: 90.0% AC: 94.3% AC: 96.8% | Nicely Visualize infected lung area to detect SARS-CoV-2 | Operational complexity is slightly higher |
| Sait et al. (2021) | Sound, Image | Inception | 4,558 | 2 | CV | T 80%: V 20% | AC: 80% AC: 99.66% | High performed Multi-modal for SARS-CoV-2 analysis | Time complexity higher and Limited data |

**Table 5 RNN method for SARS-CoV-2 analysis.**

| Author | Data type | Method | Data size | C | V | T and V size | Result | Key contribution and findings | Research gaps |
|---|---|---|---|---|---|---|---|---|---|
| Rayan & Alaerjan (2023) | X-ray image | BiLSTM | 161 | 3 | CV | T 80%: V 20% | AC 93.47%, S: 96.15%, SP 90% | Enhancing Crow Search Optimization through Bi-LSTM Model for SARS-CoV-2 Infection Identification and Classification | Worked with very small dataset |
| AlMohimeed et al. (2023) | X-ray image | Stacked RNN | 286, 4,347 | 2 | CV | T 80%: V 20% | AC: 96.83% 98.28% | Utilized stacked ensemble model using SARS-CoV-2 symptoms and chest X-ray images for the detection of the disease. | Testing performance is lower comparatively |
| Aslan et al. (2021) | X-ray image | Stacked RNN | 2,905 | 3 | CV | T 80%: V 20% | AC: 98.14% 98.70% | BiLSTM used to handle temporal properties of SARS-CoV-2 image and CNN used for the classification. | Needs to be improve to increase the adaptability to handle new data for automatic detection of SARS-CoV-2. |
| Muhammad et al. (2022) | X-ray image | CNN BiLSTM | 900, 1,212, 2,020, 2,232 | 2 | CV | T 80%: V 20% | AC: 97%, 84%, 98% | CNN extracts high-level features from the pooling layer, the augmentation mechanism selects relevant features for low dimensional augmentation, and BiLSTM is employed to classify the processed sequential information. | Feature extraction and its result visualization needs to be improved. |
| Demir (2021) | X-ray image | Deep LSTM | 761 | 3 | CV | T 80%: V 20% | AC: 100% | GRU to extract features from the chest X-ray images, and then uses a CNN layer to classify | Time and space complexity is higher |
| Afshar et al. (2020) | X ray | RNN Capsule Network | 13,800 | 3 | CV | T 90%: V 10% | AC: 95.7%% P: 95.8% SN: 90% | Hybrid mechanism of capsule and RNN gives good outcome | Less adaptable with big data |
| Islam et al. (2020) | X ray | CNN- LSTM | 40,000 | 3 | CV | T 80%: V 20% | AC: 99.4% S: 99.2% F1: 98.9% | Hybrid but nicely handle features and classify it | Need to be adaptable to analyse multi class data |
| Ozturk et al. (2020) | X-ray | DarkCovidNet | 1,000 | 2 | 5 Fold | T 80%: V 20% | AC: 98.08%, P: 98.03%, R: 95.13%, F1: 96.51% | Relatively more successful in detection of SARS-CoV-2 | Less Robust Did not handle real-time data |
| Shah et al. (2021) | X-ray | CNN-BiGRU | 424 | 3 | CV | T 80%: V 20% | AC: 96.00%, P: 96.00%, R: 96%, F1: 95% | Relatively successful as a hybrid model in SARS-CoV-2 detection | Data augmentation problem |

**Table 6  Capsule network method for SARS-CoV-2 analysis.**

| Author | Data type | Method | Data size | C | V | T and V size | Result | Key contribution and findings | Research gaps |
|---|---|---|---|---|---|---|---|---|---|
| *Yuan et al. (2023)* | X-ray image | Lightweight capsule network | 546, 2,686 | 3 | CV | T 80%: V 20% | AC: 97.99%, P: 98.05%, R: 98.02%, F1: 98.03% | New feature extractor successfully captures SARS-CoV-2 pathological feature using depthwise, point, and dilated convolution, while constructing a classification layer using homogeneous vector capsules. | Clinical research and testing are necessary. |
| *Sharma et al. (2023)* | X-ray image | Convolutional capsule network | 15,153 | 3 | CV | T 80%: V 20% | AC: 97.69%% | Capsule network handles spatial information with convolutional layers for efficient feature extraction that increase the classification performance. | Implying that the majority class is overfitted by the model. |
| *Wen et al. (2023)* | X-ray image | Attentive capsule network | 23,409 | 3 | CV | T 80%: V 20% | AC: 96.3%, S: 98.8%, SP: 93.8%, ROC: 98.3% | Study proposes attention capsule sampling network for SARS-CoV-2 detection using key slices enhancement method on chest CT scans. | Inadequate feature extraction for individual slices or patient clinical information. |
| *Zhao et al. (2023)* | X-ray image | Deep convolutional correlation capsule network | 1,025 | 3 | CV | T 80%: V 20% | AC: 100% 94.38% 86.36%. | Correlative feature extraction uses multi-level capsules with primary, correlative, and digit capsules to learn correlations and select the best digital capsules. | Limited labelled images, monotonous models, data cause biased learning, inaccurate auxiliary diagnosis. |
| *Li et al. (2023)* | X-ray image | ResNet capsule | 4,236, 2,250 | 2 | CV | T 80%: V 20% | AC: 99.88% 99.33% | ResCapsNet Features are extracted using an enhanced Residual feature extraction network, followed by the utilization of multiple Capsule Networks for SARS-CoV-2 and Non-SARS-CoV-2 classification. | Weak to demonstrate the generalization. |
| *Afshar et al. (2020)* | X ray | Capsule network | 13,800 | 3 | CV | T 90%: V 10% | AC: 95.7%% P: 95.8% SN: 90% | Hybrid mechanism of capsule and RNN gives good outcome | Less adaptable with big data |
| *Mobiny et al. (2020)* | CT scan | Capsule network | 746 | 2 | CV | T 80%: V 20% | AC: 87.6% P: 84.3% S: 85.2% F1: 87.1% | Sufficiently handle feature | Used very small data |

**Table 7 Attention network method for SARS-CoV-2 analysis.**

| Author | Data type | Method | Data size | C | V | T and V size | Result | Key contribution and findings | Research gaps |
|---|---|---|---|---|---|---|---|---|---|
| *Ullah et al. (2023)* | X-rays image | Densely attention network | 17,342 | 2 | CV | T 80%: V 20% | AC: 97.22%, S: 96.87%, SP: 99.12%, P: 95.54% | Dense layers extract spatial features, channel attention builds weights, suppresses redundant representations. | Need to Expand model for pneumonia and other lung diseases to aid radio-logists. |
| *Ouyang et al. (2020)* | CTs image | Dual-sampling attention network | 3,774 | 2 | CV | T 80%: V 20% | AC: 87.5%, S: 86.9%, SP: 90.1%, F1: 82.0%. | Divide the areas affected by pneumonia infections to increase network focus and improve visual attention for more comprehensible and interpretable models. | Prediction accuracy should be adaptive with large data |
| *Yang et al. (2023)* | X-ray image | Attention-based transformer | 15,153 | 2 | 5K-CV | T 80%: V 20% | AC: 98.0% | Feature extraction and classification with Integrated Attention-based transformers model outperforms CNN in SARS-CoV-2 diagnosis. | Computational complexity |
| *Wen et al. (2023)* | X-ray image | Attentive capsule network | 23,409 | 3 | CV | T 80%: V 20% | AC: 96.3%, S: 98.8%, SP: 93.8%, ROC: 98.3% | Study proposes attention capsule sampling network for SARS-CoV-2 detection using key slices enhancement method on chest CT scans. | Inadequate feature extraction for individual slices or patient clinical information. |
| *Christina Magneta, Sundar & Thanabal (2023)* | X-ray image | Hierarchical attention network | NA | 2 | CV | T 80%: V 20% | AC: 93.36% | MRMVO-based HAN classifier incorporates manta-ray foraging optimization and multi-verse optimizer, acquiring SARS-CoV-2 detection features from segmented lung lobes for targeted regions. | Time complexity need to be decreased |
| *Hu et al. (2022)* | X ray | Attention ResNet | 4,449 | 3 | CV | T 80%: V 20% | SN: 90.2% | Attention fusion feature for automatic SARS-CoV-2 from CT images | Result is biased to image segmentation |
| *Fan et al. (2021)* | X-ray image | Multi-kernel-size spatial channel attention | 1,000 | 2 | 10K CV | T 80%: V 20% | AC: 98.2% | First stage extracts features, second stage uses multi-kernel attention modules, segmented pneumonia infection regions, for improved model interpretability and explainability. | Data modality and model generability should be improved |
| *Zhang et al. (2021)* | X-ray | Attention GAN | 100 | 2 | CV | T 80%: V 20% | S: 69.85% P: 94.6% | Hybrid model that performs good with some prepossessing | Worked on small data |
| *Wang et al. (2021b)* | CT scan | Attentive FGCN | 320 | 2 | CV | T 80%: V 20% | AC: 97.71% | Handles feature context efficiently | It is not adaptable enough |

**Table 8 Hybrid method for SARS-CoV-2 analysis.**

| Author | Data type | Method | Data size | C | V | T and V size | Result | Key contribution and findings | Research gaps |
|---|---|---|---|---|---|---|---|---|---|
| *Sejuti & Islam (2023)* | CTs image | CNN–KNN | 4,085 | 2 | 5K-CV | T 80%: V 20% | AC: 98.26%, P: 99.42%, R: 97.2%, F1: 98.19% | Uses less paprameters of CNN and KNN that detects SARS-CoV-2 with good generability and less ovierfitting problem. | Performance is bound to small dataset |
| *Basha et al. (2023)* | CTs image | Hybrid with neurosymbolic | 1,885 | 2 | CV | T 80%: V 20% | AC: 98.7%, S: 99.8%, SP: 96.4% F1: 99.05% | Two experiments investigate automated SARS-CoV-2 detection using NRCS and genetic-based methods, using neurotrophic logic. | Time complexity should be reduced |
| *Pustokhin et al. (2023)* | X-ray image | Attention BiLSTM | 646 | 5 | CV | T 80%: V 20% | S: 93.28%, SP: 94.61%, P: 94.90%, AC: 94.88%, F1: 93.10% | RCAL-BiLSTM model uses bilateral filtering, feature extraction, and softmax classification for image classification. | Real time implementation should be developed |
| *Deepak et al. (2023)* | X-ray image | Quantum neural network with RFNN | 3,871 | 4 | CV | T 80%: V 20% | AC: 99.25% | Hybrid median bilateral filter, SC-ResNet 50 segmentation, robust feature neural network extraction, DSFSAM, HWOA, and deep-QNN classify X-rays into multiple disease classes, reducing noise and enhancing infected regions. | Study should explore optimization techniques for system loss reduction. |
| *Hu et al. (2022)* | X ray | Attention ResrNet | 4,449 | 3 | CV | T 80%: V 20% | SN: 90.2% | Attention fusion feature for automatic SARS-CoV-2 from CT images | Result is biased to image segmentation |
| *Ismael & Şengür (2021)* | X-ray | CNN, SVM | 380 | 2 | CV | T 80%: V 20% | AC: 94.7% | Use SVM and CNN detect SARS-CoV-2 fastly | Weak performance limited data |
| *Zhang et al. (2021)* | X-ray | Attention GAN | 100 | 2 | CV | T 80%: V 20% | S: 69.85% P: 94.6% | Hybrid model that performs good with some prepossessing | Small data |
| *Wang et al. (2021b)* | CT Scan | FGCN | 320 | 2 | CV | T 80%: V 20% | AC:97.71% | Handles feature context efficiently | It is not adaptable enough |
| *Islam et al. (2020)* | X ray | CNN-LSTM | 40,000 | 3 | CV | T 80%: V 20% | AC: 99.4% S: 99.2% F1: 98.9% | Hybrid but nicely handle features and classify it | Need to be adaptable to analyse multi class data |
| *Shah et al. (2021)* | X-ray | CNN-BiGRU | 424 | 3 | CV | T 80%: V 20% | AC: 96.00%, P: 96.00%, R: 96%, F1: 95% | Relatively successful as a hybrid model in SARS-CoV-2 detection | Data augmentation problem |

Figures 12 and 13 depict graphical representation of performance (training as well as validation with loss and accuracy) for twelve models: CNN, VGG16, ResNet50, InceptionNetV3, NasNetMobile, DenseNet121, XceptionNet, AlexNet, CNN-RNN, EfficientNetB2, and MobileNetV2. Sub caption indicates the name of each model implemented. As shown in Fig. 13, loss and accuracy curves of those deep learning models offer valuable insights into their performance during training. This loss curve charts the model's ability to minimize the error between predicted and actual outputs over time, showcasing improved learning and convergence through a declining trend. Conversely, the accuracy curve gauges the model's success in correctly classifying inputs, illustrating its

**Table 9 Performance analysis (COVID-19 image data).** The bold text with the value indicates better performance, characterized by higher accuracy and lower loss.

| Model | Total parameter | Train accuracy | Train loss | Validation accuracy | Validation loss |
|---|---|---|---|---|---|
| CNN | 9,014,660 | 0.9761 | **0.0367** | 0.9400 | 0.09550 |
| VGG16 | 138,357,544 | 0.9768 | 0.0467 | **0.9420** | 0.09440 |
| ResNet | 25,636,712 | 0.7532 | 0.3064 | 0.7987 | 0.2749 |
| InceptionNetV3 | 23,851,784 | 0.9707 | 0.0404 | 0.6175 | 0.1074 |
| DenseNet121 | 8,062,504 | 0.8101 | 0.2739 | 0.6175 | 0.3904 |
| XceptionNet | 22,910,480 | 0.86623 | 0.1756 | 0.8800 | 0.1645 |
| AlexNet | 62,300,000 | 0.8968 | 0.1264 | 0.8737 | 0.1613 |
| CNN-RNN | N/A | 0.8663 | 0.2715 | 0.8150 | 0.3245 |
| EfficientNetB2 | 9,177,569 | 0.9707 | 0.0904 | 0.2475 | 5.5966 |
| MobileNetV2 | 3,538,984 | **0.9797** | 0.0402 | 0.6713 | 0.6609 |
| VGG-19 | 143,667,240 | 0.9820 | 0.0461 | **0.9410** | **0.0951** |
| MobileNet-V3 | 5,481,752 | **0.9811** | 0.0401 | 0.6712 | 0.6601 |

predictive accuracy. Monitoring these curves throughout training enables the assessment of model behaviour, the identification of convergence issues or overfitting, and the adjustment of training parameters to enhance overall performance. Together, these curves serve as crucial diagnostic tools for understanding and optimizing the performance of deep learning models. As shown in Figs. 12 and 13, there are some overfitting and underfitting problems in the performance of DenseNet, EfficientNet and CNN-RNN models. But other model performs well without fitting issues.

Table 9 presents the SARS-CoV-2 crucial outcome analysis. The best model is VGG-19, which has the highest training accuracy of 98.20%, while the MobileNet-v3 technique performs similarly to MobileNet, with 98.11 percent accuracy and less training loss of 0.0401. In terms of validation performance, VGG-16 outperforms the others with a validation accuracy of 94.20% and a validation loss of 0.09440. MobileNetV3 is suggested for training while VGG16 is suitable for validation activities.

# DISCUSSION AND RECOMMENDATION

The application of deep learning algorithms has exhibited remarkable progress and significant advancements in both image-based and speech-based SARS-CoV-2 analysis within the realm of deep learning research, as explored in this article. Notably, among the studied methodologies, RNN-based approaches exhibit relatively inferior performance compared to CNN and its hybrid counterparts. Nonetheless, RNN-based algorithms, particularly when coupled with attention mechanisms, capsule networks, and autoencoder-decoder structures, display promise in the domains of image-based SARS-CoV-2 analysis and prediction.

In the context of the ongoing SARS-CoV-2 pandemic, the prioritization of health policy considerations over potential privacy concerns gains prominence. While delving into SARS-CoV-2 analysis utilizing image data, it is imperative to meticulously address issues

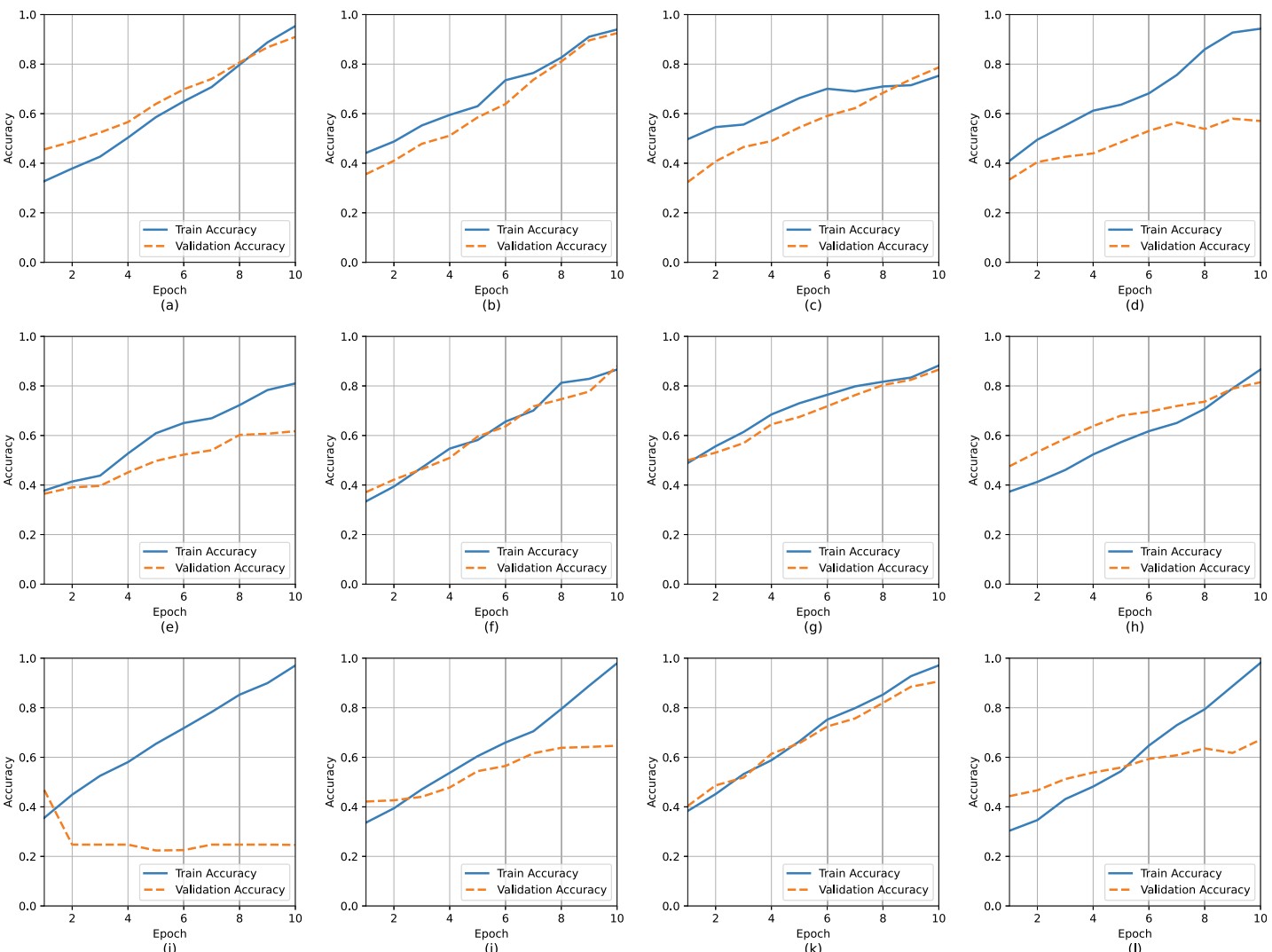

**Figure 12 Accuracy curve during training and validation: (A) CNN, (B) VGG-16, (C) ResNet, (D) InceptionNetV3, (E) DenseNet121, (F) XceptionNet, (G) Alexnet, (H) CNN-RNN, (I) EfficientNetB2, (J) MobileNetV2, (K) VGG-19, (L) MobileNetV3.**

of transparency, privacy concerns, human safety, and the potential misuse of data. Ethical considerations, governmental regulations, political laws, and privacy safeguards should all play a crucial role in medical data analysis.

This study also explores potential avenues for future research and highlights challenges associated with auxiliary SARS-CoV-2 datasets. The opacity surrounding data and analytic processes poses a significant hurdle for data-driven AI initiatives. Enhancing computed tomography and X-ray datasets can substantially bolster the precision and efficacy of deep learning algorithms in image-based analyses.

Undoubtedly, image-based data holds a wealth of critical information for the detection of SARS-CoV-2. While many models are yet to undergo real-world testing, their potential

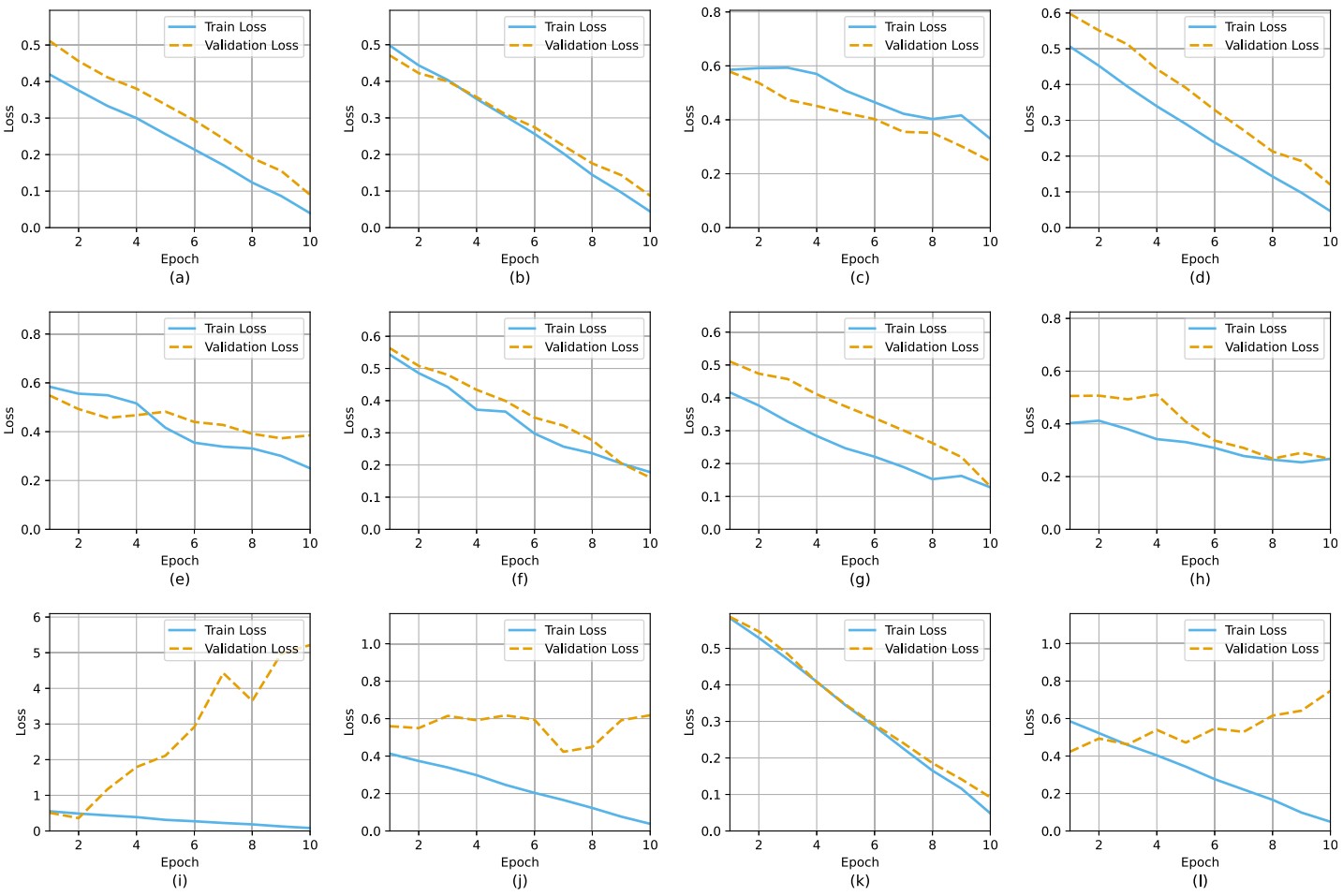

**Figure 13 Loss curve during training and validation: (A) CNN, (B) VGG-16, (C) ResNet, (D) InceptionNetV3, (E) DenseNet121, (F) XceptionNet, (G) Alexnet, (H) CNN-RNN, (I) EfficientNetB2, (J) MobileNetV2, (K) VGG-19, (L) MobileNetV3.**

contribution to mitigating the SARS-CoV-2 outbreak remains substantial. Nonetheless, the study itself acknowledges certain limitations. The specifics of neural network architectures, including the number of layers, layer configurations, learning rates, intervals, batch sizes, dropout layers, optimization techniques, and transfer functions, are not explicitly outlined. Additionally, while the study approaches SARS-CoV-2 diagnosis from a computational standpoint, it lacks qualitative diagnostic indications within CT or X-ray image data.

Deep learning methodologies, although powerful, necessitate a substantial volume of labelled data to ensure accurate predictions. The process of collecting and labelling such extensive datasets can be arduous and time-consuming. The opaque nature of deep learning models underscores the challenge of ensuring deep machine outputs align with desired expectations. Incorporating domain knowledge can facilitate the fine-tuning of model parameters, enhancing system reliability.

**Table 10 Drawback of different deep learning models for SARS-CoV-2 analysis from the image.**

| Model | Limitations | Potential solutions |
|---|---|---|
| CNN | Slow computation (*Islam & Matin, 2020*), Need more parametric initialization (*Yu et al., 2021*), Can not encode object rotation (*Islam, Islam & Asraf, 2020*). | Ability to automatically extract and classify intricate features in medical images, such as chest X-rays and CT scans, with high accuracy and efficiency (*Bhattacharyya et al., 2022*). |
| RNN | Learning is slow with Vanishing of the gradient (*Imran et al., 2020*), Unable to handle complex features of image (*Raamkumar, Tan & Wee, 2020*). | Effective for COVID-19 detection and analysis by processing and analyzing sequential medical data to identify long-range patterns indicative of the virus over time (*Hassan, Shahin & Alsabek, 2020*). |
| LSTM | Space complexity is higher, Overfitting problem (*Pahar et al., 2021*). | Good outcome in COVID-19 detection and analysis by accurately capturing long-term dependencies and patterns in sequential medical data with facilitating reliable prediction. |
| GRU | Slow convergence rate (*Wang et al., 2020c*), Low training efficiency, High time complexity (*Jiang et al., 2020*), Under-fitting problem (*Wang et al., 2020c*). | Efficiently capturing temporal dependencies and patterns in sequential medical data that enable accurate prediction and diagnosis of the virus (*Jiang et al., 2020*). |
| CNN-RNN | Low performance when handling small objects with fewer features (*Islam, Islam & Asraf, 2020*; *Ardakani et al., 2020*). | Integrating image features extracted by CNNs with sequential data analysis by LSTMs, providing a robust and accurate approach (*Ardakani et al., 2020*). |
| Attention | Higher parameters (*Paka et al., 2021*), Increased time complexity (*Pinkas et al., 2020*). | Enhance model focusing on relevant features within medical images or sequential data, improving accuracy by prioritizing critical features indicative of the virus (*Paka et al., 2021*). |
| Capsule | Not speedy in computation (*Afshar et al., 2020*), Higher time and space complexity (*Yuan et al., 2023*) than CNN. | Efficiently captures hierarchical relationships in image features, enabling precise identification of subtle virus-related patterns with improved generalization (*Afshar et al., 2020*). |
| Hybrid | Complex (*Sejuti & Islam, 2023*) and costly in operation (*Mobiny et al., 2020*). | Deliver high performance by leveraging the complementary strengths of multiple architectures, sequential data processing and focusing on relevant features, thereby enhancing diagnostic accuracy and robustness (*Basha et al., 2023*). |

Technical aspects such as time and space complexity, reliance on high-performance hardware like GPUs, and substantial RAM requirements underscore the demands of deep learning approaches. Despite limited real-world testing, many models exhibit potential in addressing the SARS-CoV-2 pandemic. Future researchers are encouraged to focus on problem resolution, including addressing contemporary challenges. Additionally, the study highlights the rarity of multimodal SARS-CoV-2 analysis and suggests it as a potential avenue for further exploration.

Based on our analytical examination of twelve advanced deep learning models, MobileNet is recommended for training, while VGG16 is advised for validation tasks. The strengths and limitations of various deep learning techniques for SARS-CoV-2 analysis from image data are succinctly presented in Table 9. Furthermore, Table 10 provides a comprehensive overview of the limitations associated with several deep learning methodologies in the context of SARS-CoV-2 research.

## Potential of the proposed method across different demographics and healthcare settings

The potential application of deep learning techniques for SARS-CoV-2 detection using X-ray and CT images across diverse demographics and healthcare settings offers significant

promise. These methods can improve diagnostic accuracy for varied populations by using diverse, representative datasets. Enhancements like transfer learning and domain adaptation will boost model robustness, ensuring reliable performance with different imaging protocols and equipment. Optimizing models for low-resource environments and implementing cloud-based solutions will increase global accessibility to advanced diagnostic tools. Adhering to ethical guidelines and ensuring robust data privacy will be essential for maintaining trust and regulatory compliance.

Beyond COVID-19, these deep-learning techniques can be adapted to diagnose other respiratory diseases, aid early detection of emerging infectious diseases, and monitor disease progression. They can also be integrated into telemedicine platforms to provide remote diagnostic support, especially in underserved regions. Regular updates to the models, along with comprehensive training for healthcare providers, will keep these AI-driven solutions accurate and effective, ultimately enhancing global healthcare response capabilities and improving patient outcomes across various settings.

## Limitation of deep learning model

The application of deep learning models in SARS-CoV-2 image analysis is not without its limitations, encompassing both technological and mathematical constraints. Several limitations are associated with deep learning techniques including:

1. Data quantity prerequisites for training (*Subramaniam et al., 2023*): Achieving significant performance gains over alternative approaches demands a substantial volume of data. Deep learning thrives on extensive datasets for effective training.

2. Computational and space complexity (*Asif et al., 2023*): Time and space complexity issues in SARS-CoV-2 detection using X-ray image deep learning are multifaceted. Training's extensive duration due to complex architectures and large datasets delays model development. Deployment encounters high inference time, hampering real-time diagnosis in clinical settings. Resource-intensive computations require robust hardware, limiting deployment in resource-constrained environments. Large model and dataset sizes raise storage and infrastructure costs. Complex model interpretation hinders understanding of prediction rationale. Balancing overfitting prevention and predictive accuracy is delicate. Ensuring model generalization across diverse conditions adds complexity. Addressing these challenges is vital to harnessing deep learning's potential for effective SARS-CoV-2 detection *via* X-ray images.

3. Interpretability challenges (*De Falco, De Pietro & Sannino, 2023*; *Islam et al., 2023b*): Deciphering output solely based on training is intricate, often requiring the implementation of classifiers. Techniques rooted in convolutional neural networks are commonly employed to address this challenge.

4. Black box problems (*De Falco, De Pietro & Sannino, 2023*): Deep learning models learn complex patterns from extensive image datasets that are not easily discernible by humans. However, their opaque decision-making process, known as the black box problem, creates trust, refinement, and application challenges, especially in clinical settings. Strategies like interpretable machine learning, model distillation, adversarial

training, self-explaining neural networks, and visualization techniques aim to address this issue. Ongoing research seeks to enhance deep learning model reliability and usefulness, particularly in clinical contexts. The black box problem is particularly significant in deep learning due to its reliance on concealed patterns in large image datasets. This opacity undermines confidence in predictions, vital in healthcare where understanding model operations is key for informed decisions. Debugging and improving models are complicated by this problem, as identifying issues without comprehending internal workings hampers performance enhancements.

5. Overfitting problems (*Meedeniya et al., 2022*): Overfitting is a concern in machine learning, including deep learning for SARS-CoV-2 detection *via* X-ray images. It happens when a model becomes overly attuned to training data, hindering its ability to generalize to new data. Consequently, this can result in inaccurate predictions. Overfitting arises from factors like a small training dataset, limiting the model's exposure to genuine patterns. Excessive model complexity exacerbates this issue by causing the model to memorize data rather than learn meaningful insights. Regularization, a technique to prevent overfitting, introduces a penalty to the model's loss function, promoting simplicity and curbing the model's tendency to grasp irrelevant training data patterns.

6. Lack of contextual understanding (*Hu et al., 2022*): Deep learning models used for SARS-CoV-2 detection from X-ray images learn patterns associated with the disease but lack contextual understanding, leading to potential inaccuracies in predictions and challenges in explanation. For instance, the model might identify SARS-CoV-2-related features without comprehending their significance within the overall image. This hinders its ability to explain predictions and poses trust issues in clinical settings. To tackle this limitation multimodal data, transfer learning methods and interpretable machine learning can be employed. Real-time SARS-CoV-2 data problems:

7. Real-life data problems (*Srivastava et al., 2022*): Real time data helps to detect and predict disease precisely. In SARS-CoV-2 detection using X-ray image deep learning includes the following real time data problems:

(a) Data scarcity (*Ali, Grönlund & Shah, 2023*): Limited SARS-CoV-2 data due to the disease's novelty hampers accurate model training, reducing comparability with diseases possessing more data; (b) Data imbalance (*Calderon-Ramirez et al., 2021*; *Chamseddine et al., 2022*): Unequal representation of SARS-CoV-2 cases *vs* non-SARS-CoV-2 cases makes it challenging for models to correctly identify the disease; (c) Data Noise (*Momeny et al., 2021*): Noisy data containing artifacts interferes with model learning, impacting accuracy; (d) Data bias (*Catalá et al., 2021*): Biased data not representing diverse populations reduces model accuracy for certain groups.

These drawbacks compound other issues such as complex model training, lack of robustness to real-world variations, and computational resource demands during deployment. Despite these hurdles, addressing these problems through research efforts can lead to improved deep-learning models for early SARS-CoV-2 detection and diagnosis using X-ray images.

### Drawbacks of different deep learning models

Different deep learning models for COVID-19 detection from images have drawbacks (*Alaufi, Kalkatawi & Abukhodair, 2024*): CNNs lack interpretability, RNNs struggle with long-range dependencies, LSTMs are complex, GRUs may lack expressiveness, attention mechanisms introduce computational overhead, and capsule networks have limited research (*Sharma & Guleria, 2024*). A comprehensive overview of these limitations is provided in Table 10, encompassing detailed descriptions of the associated constraints for individual model in SARS-CoV-2 detection.

## Ethical considerations and privacy issues in SARS-CoV-2 analysis

Ethical concerns and validation are paramount when using patient images for AI training in COVID-19 detection. Ensuring informed consent from patients is essential, alongside robust anonymization procedures to safeguard privacy. Compliance with regulations like General Data Protection Regulation (GDPR) (*Varpula et al., 2024*; *Tertulino, Antunes & Morais, 2024*) and Health Insurance Portability and Accountability Act (HIPAA) (*Carmine, 2024*) is mandatory to ensure data protection. Validation of AI models must involve rigorous testing to ensure accuracy and reliability, with transparency in the validation process to maintain trust. Addressing potential biases in the training data is crucial to ensure fairness and equitable outcomes (*Vrudhula et al., 2024*). Ultimately, adherence to ethical guidelines and validation protocols is essential to uphold patient rights and ensure the ethical use of AI in COVID-19 detection (*Abedi et al., 2024*).

Preserving privacy in the context of SARS-CoV-2 data analysis remains a critical concern. The encryption and secure management of medical images related to SARS-CoV-2 has been suggested as an effective approach (*Durafe & Patidar, 2022*). Within this realm, the emphasis is placed on ensuring the confidentiality, accessibility, and security of SARS-CoV-2-related information (*Shuja et al., 2021*). Big data challenges intertwined with SARS-CoV-2 monitoring methods encompass issues of security, safety, and sustainability, especially pertinent in the context of pandemic-induced health concerns. Managing large datasets within data centres during SARS-CoV-2 has posed challenges, particularly in the areas of AI, deep learning, and statistical research that utilize user data from sources like medical centres and social media. The omnipresent concern for privacy continues to be a focal point, with the existence of open-source data exacerbating privacy issues. These privacy concerns can sometimes overshadow critical public health matters, potentially leading to constraints on the sharing of scientific data (*Shuja et al., 2021*).

There is a looming apprehension that post-pandemic, surveillance practices may persist beyond the realm of public health. SARS-CoV-2-related applications have facilitated data exchange with third parties, raising legitimate concerns about user privacy (*Naudé, 2020*). While patient data collection has been supported by hospitals and medical institutions, concerns persist regarding the thorough assessment of stability and privacy vulnerabilities within these datasets, despite efforts towards data anonymization. The automatic contact tracing measures adopted by governments in response to SARS-CoV-2 transmission have also prompted deliberations over consumer privacy considerations (*Chan et al., 2020*).

In the pursuit of scholarly purposes, data sharing mandates meticulous protection of both physician and mobility data to avert biases. Federated deep learning (DL)-based learning approaches offer a potential solution, as they circumvent the need to centralize data at a single cloud data centre. In federated DL, model parameters and outputs are communicated between participating nodes, ensuring data protection within a cohesive machine learning framework (*Liu et al., 2020*). Furthermore, in the context of the SARS-CoV-2 pandemic, the prioritization of public health concerns may take precedence over matters of individual privacy (*Latif et al., 2020*).

In summary, the intricate balance between data utilization for vital research and safeguarding privacy remains a central challenge in SARS-CoV-2-related data analysis.

## Challenges in SARS-CoV-2 analysis

Challenges of different types of deep learning methods are addressed here, the challenges can boost researchers to develop a new and improved method to analyse SARS-CoV-2.

1. SARS-CoV-2 data generation (*Phukan et al., 2022*): Manual data generation is challenging in the field of health-related concern. We should be assisted by health or clinical institutions to get patient data. Permission and government allowance to make available for analysis is the big challenging issues in data generation.

2. Data availability (*Tsai et al., 2021*): Most of data set for SARS-CoV-2 analysis is public and some are private. To analyse all these data sets, we have to be aware about its verification of labelling, augmentation and balancing (*Tsai et al., 2021*).

3. SARS-CoV-2 data labelling legislation and verification (*Yang et al., 2021*): SARS-CoV-2 results are exchanged according to strict regulations in countries all over the world. One of the primary procedures specifically states that the bare minimal amount of specimens and data from infected individuals must be gathered in the shortest amount of time. As a result, analysis is becoming more and more difficult to perform. Before using data to analyse, proper data labelling and its correctness need to be confirmed by the source and its released licence.

4. SARS-CoV-2 data augmentation (*Nishio et al., 2020*): SARS-CoV-2 data augmentation approaches are becoming more popular for analysing output quality. The data augmentation field demands new research and study in order to provide fresh/synthetic data with sophisticated applications. Artificial Intelligence tough to use GANs to create high-resolution images, for example. If there are biases in the original dataset, there will be biases in the data supplemented from it. As a result, figuring out the optimal data augmentation strategy is essential.

5. Noisy SARS-CoV-2 data (*Momeny et al., 2021*): The presence of noisy SARS-CoV-2 data in a data set can have a major impact on the accuracy of any useful information prediction. Much empirical research has demonstrated that data set noise significantly reduced classification accuracy and resulted in poor prediction outcomes.

6. Quality and dimension of SARS-CoV-2 image data (*Rajinikanth et al., 2020*): SARS-CoV-2 image quality should be good quality other it may lead deep learning classifier to predict wrong classification. Normally, most of the raw data from medical institutes

are in 3D form and deep learning models are not suitable to be trained with 3D data, it can deal with 2D data. So it should be a concern and solved.

7. Imbalanced SARS-CoV-2 data (*Calderon-Ramirez et al., 2021*; *Chamseddine et al., 2022*): Because of the significantly lopsided class distribution and disproportionate misclassification costs, balanced categorization is particularly difficult. Properties including dataset size, label noise, and data dispersion add to the difficulty of imbalanced classification. Resampling SARS-CoV-2 data and other tools can handle imbalanced data.

8. SARS-CoV-2 data preprocessing (*Maity, Nair & Chandra, 2020*): There are some challenges in SARS-CoV-2 dat preprocessing like: missing data, manual input, data inconsistency, regional formats, wrong data types, file manipulation, and missing anonymization. Data preprocessing with data cleaning, data integration, data transformation, and data reduction are all examples of data preprocessing. Data cleaning is a technique for removing noise and correcting data discrepancies.

9. Model development, improvement (*Chauhan, Palivela & Tiwari, 2021*) and tuning (*Lee et al., 2020*) for SARS-CoV-2 detection: Because of the significantly lopsided SARS-CoV-2 class distribution and disproportionate misclassification costs, balanced categorization is particularly difficult. Properties including dataset size, label noise, and data dispersion add to the difficulty of imbalanced classification. Deep learning models have challenges if there are the following issues like: not enough training data, poor quality of data, irrelevant features, non-representative training data, overfitting and underfitting.

10. Intensive training expenses (*Gupta et al., 2023*): The complexity of SARS-CoV-2 data models results in resource-intensive training processes. The computational requirements necessitate the use of numerous workstations and costly GPUs, leading to elevated operational expenses.

11. Skill and expertise demand (*Attallah, 2023*): Deep learning lacks a standardized model selection process. Optimal selection demands an understanding of topology, training methodologies, and other intricate characteristics, often posing challenges for individuals with limited expertise.

## Future scopes

A plethora of deep learning model implementations has been applied across diverse datasets, with a particular emphasis on radiology imaging datasets, as showcased and explored in prior research (*Liu et al., 2020*; *Nishio et al., 2020*). However, the translation of these techniques into real-world patient care scenarios remains challenging, underscoring the pressing necessity for benchmarking frameworks that can assess and compare novel methodologies. Such frameworks can facilitate the practical deployment of analytical hardware infrastructures by accounting for common medical data, data preprocessing procedures, and assessment requirements across various AI approaches, thereby ensuring data transparency and interpretability (*Tsai et al., 2021*; *Phukan et al., 2022*).

In today's world, advancements in sophisticated technologies like artificial intelligence are pervasive and essential in combating SARS-CoV-2. Integrating AI can not only offer intrinsic advantages but also mitigate misinformation dissemination on social media platforms (*Bharati et al., 2021*). By analyzing vast amounts of data, AI ensures the dissemination of accurate information, preempting premature panic and hysteria amidst the pandemic.

Cross-disciplinary collaboration is paramount in advancing the field of COVID-19 detection, as it enables researchers to gain insights into clinical needs and challenges from diverse perspectives (*Ritto et al., 2024*). This collaboration between computer scientists, medical professionals, and public health experts ensures the development of AI models that will be effective and clinically relevant. Moreover, the development of accessible models is crucial for widespread adoption in healthcare settings. Future research should prioritize user-friendly interfaces, comprehensive training programs, and compatibility with existing healthcare systems to facilitate seamless integration. Additionally, achieving generalizability across different demographics, regions, and healthcare settings is essential (*Abad, Casas-Roma & Prados, 2024*). Transfer learning techniques and domain adaptation methods can enhance model performance and adaptability. Emphasizing the interpretability of AI models is vital for building trust among healthcare professionals. Interpretable models provide clear explanations for predictions, enabling clinicians to understand and trust the AI-driven diagnostic process (*de Sousa Freire et al., 2024*). Furthermore, real-time detection techniques should be explored to enable rapid diagnosis and treatment decisions. Optimizing model architectures for speed and efficiency is crucial for achieving real-time performance (*Kaur et al., 2024*). Models should also be robust to variations in imaging quality, patient demographics, and disease severity to maintain diagnostic accuracy in real-world scenarios (*Khero, Usman & Fong, 2024*).

The integration of various types of medical and serological clinical data significantly enhances the diagnostic accuracy and robustness of models for COVID-19 detection. By providing a more comprehensive view of the patient's health, this approach helps identify the virus more reliably, thereby reducing the likelihood of false positives and negatives. This comprehensive integration ultimately supports more precise and effective patient care (*Lei & Mohan, 2020*; *Kamari et al., 2024*).

Ethical considerations are paramount in the development and deployment of AI models to ensure patient privacy, informed consent, and equity in healthcare delivery. By integrating ethical practices, researchers can contribute to the responsible use of technology in healthcare and mitigate potential biases or unintended consequences (*Bartenschlager et al., 2024*).

## CONCLUSION

The deep learning algorithm, in addition to other methods, is employed to identify and detect SARS-CoV-2. The ongoing SARS-CoV-2 pandemic continues to establish new benchmarks in terms of global disease prevalence and mortality rates, both in cumulative and daily contexts. In this scenario, an automated and reliable deep learning-based diagnosis for SARS-CoV-2 assumes paramount importance, aiding in the swift and

accurate identification of the disease. Presently, digital image assessment for SARS-CoV-2 diagnosis utilizing deep learning techniques, particularly employing two distinct imaging modalities—CT and X-ray samples, surpasses the performance of speech and text-based analysis. Image-based analysis may overlook intricate image predictions, whereas speech-based analysis can be susceptible to data noise.

Attaining superior efficacy necessitates precise labelling of image data and facilitating enhanced learning. The article delineates tools for diagnosing and analyzing SARS-CoV-2, encompassing a pre-trained transfer learning paradigm and a tailored deep learning architecture. To explore the potential of deep learning algorithms and imaging methodologies, a three-tier taxonomy is established. This review article juxtaposes various advanced deep learning techniques against the latest models, delving into operational specifics, mathematical relationships, outcomes, applications, advantages, performance metrics, and optimal model recommendations. The study also elucidates the sources of all utilized datasets, along with their associated challenges, presented in a manner conducive to comprehension and application within the scientific community.

- Implications of this research:
Through this research and analysis of SARS-CoV-2 detection with deep learning methods has broad implications. This research suggests that deep learning analysis is crucial for accurate and versatile detection, addressing knowledge gaps, benchmarking performance, and improving clinical applications. It guides how the DL method reduces workload, accelerates medical decisions, aids remote diagnosis, and presents ethical AI use and data privacy. It also enhances public awareness of AI's healthcare role. Overall, deep learning analysis drives research, informs practice, and aids pandemic control. Furthermore, the study delves into highly performing models and proposes potential enhancements to current deep learning models, thereby inviting academic engagement in this domain. It is imperative to recognize that image-based deep learning systems offer limited insights into afflicted individuals. However, this assertion does not necessarily imply that deep learning algorithms can supplant the role of physicians or clinicians in clinical diagnosis. The lack of a standardized benchmark, data balance, and accurate labelling represents a notable drawback in SARS-CoV-2 deep learning diagnostic systems. In the near future, deep learning researchers are anticipated to collaborate closely with radiologists and medical specialists, forging efficient support systems for detecting SARS-CoV-2 infections, especially during initial diagnosis and determination of infection severity. The collaboration between deep learning experts and medical professionals is projected to yield substantial progress in identifying SARS-CoV-2-infected patients. For robust practice, image-based data analysis of SARS-CoV-2 demands ample volume and meticulous labelling. Prospective endeavours encompass real-time data processing, multimodal analysis, and predictive assistance for medical practitioners. A more comprehensive analysis involving diverse models on distinct image datasets is envisioned in forthcoming investigations. This will yield clearer insights into the performance and recommendations of the most adept models. Deep learning specialists, in collaboration with radiologists, will play a pivotal role in

identifying SARS-CoV-2 infections. This review underscores the potential for building adaptive and high-performing SARS-CoV-2 analysis systems through multimodal approaches, particularly integrating image-based data analysis.

## APPENDIX

The authors utilised a lot of terminology in this lengthy review. In this appendix, we added a complete Table 11 of a comprehensive list of abbreviations to make clear comprehension and introduction to different types of phrases.

**Table 11 Comprehensive list of abbreviations used in this article.**

| Abbreviation | Definition | Abbreviation | Definition | Abbreviation | Definition |
|---|---|---|---|---|---|
| ANN | Artificial neural network | GAN | Genrative adversarial network | GCN | Graph convolutional neural network |
| ML | Machine learning | GRU | Gated recurrent unit | SVM | Support vector machine |
| DL | Deep learning | LSTM | Long short term memory | WHO | World health care organization |
| HDL | Hybrid deep learning | BiGRU | Bidirectional gated recurrent unit | ROC | Receiver operating characteristic |
| RNN | Recurrent neural network | BiLSTM | Bidirectional long short term memory | NPI | National provider identifier |
| CNN | Convolutional neural network | MLP | Multi layer perception | BOW | Bag of words |
| DNN | Deep neural network | MRI | Magnetic resonance imaging | MFC | Mel-frequency cepstrum |
| Caps | Neural network | NN | Neural network | GGO | GroundGlass opacity |
| Att | Attention-based neural network | NLP | Natural language processing | CL | Computational linguistic |
| SARS-CoV-2 | Corona virus disease 19 | CT | Computerized tomography | AI | Artificial intelligence |

### Funding

This work was supported by Multimedia University, Malaysia, Grant No. MMUI/230023. There was no additional external funding received for this study. The funders had no role in study design, data collection and analysis, decision to publish, or preparation of the manuscript.

### Grant Disclosures

The following grant information was disclosed by the authors:
Multimedia University, Malaysia: MMUI/230023.

### Competing Interests

The authors declare that they have no competing interests related to this article.

## Author Contributions

- Md Shofiqul Islam conceived and designed the experiments, performed the experiments, analyzed the data, performed the computation work, prepared figures and/or tables, authored or reviewed drafts of the article, and approved the final draft.
- Fahmid Al Farid conceived and designed the experiments, performed the experiments, analyzed the data, performed the computation work, authored or reviewed drafts of the article, and approved the final draft.
- F. M. Javed Mehedi Shamrat conceived and designed the experiments, performed the experiments, prepared figures and/or tables, and approved the final draft.
- Md Nahidul Islam conceived and designed the experiments, analyzed the data, performed the computation work, prepared figures and/or tables, authored or reviewed drafts of the article, and approved the final draft.
- Mamunur Rashid conceived and designed the experiments, performed the experiments, analyzed the data, performed the computation work, prepared figures and/or tables, authored or reviewed drafts of the article, and approved the final draft.
- Bifta Sama Bari conceived and designed the experiments, prepared figures and/or tables, authored or reviewed drafts of the article, and approved the final draft.
- Junaidi Abdullah conceived and designed the experiments, authored or reviewed drafts of the article, and approved the final draft.
- Muhammad Nazrul Islam conceived and designed the experiments, authored or reviewed drafts of the article, and approved the final draft.
- Md Akhtaruzzaman conceived and designed the experiments, authored or reviewed drafts of the article, and approved the final draft.
- Muhammad Nomani Kabir conceived and designed the experiments, authored or reviewed drafts of the article, and approved the final draft.
- Sarina Mansor conceived and designed the experiments, authored or reviewed drafts of the article, and approved the final draft.
- Hezerul Abdul Karim conceived and designed the experiments, authored or reviewed drafts of the article, and approved the final draft.

## Data Availability

This is a literature review.

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
