# Peer review of "Challenges issues and future recommendations of deep learning techniques for SARS-CoV-2 detection utilising X-ray and CT images: a comprehensive review"

_PeerJ Computer Science, doi:10.7717/peerj-cs.2517_

## Round 0.1 · original submission · Major Revisions

The reviewers have substantial concerns about this manuscript. The authors should provide point-to-point responses to address all the concerns and provide a revised manuscript with the revised parts being marked in different color.

Reviewer 1 has suggested that you cite specific references. You are welcome to add it/them if you believe they are relevant. However, you are not required to include these citations, and if you do not include them, this will not influence my decision.

**Language Note:** The review process has identified that the English language must be improved. PeerJ can provide language editing services - please contact us at [email protected] for pricing (be sure to provide your manuscript number and title). Alternatively, you should make your own arrangements to improve the language quality and provide details in your response letter. – PeerJ Staff

Reviewer 1 ·

Basic reporting

Overall, the manuscript is well-written and addresses an important topic in the field of deep learning and medical image analysis. With some revisions, it has the potential to make a valuable contribution to the SARS-CoV-2 detection and COVID-19 management.

Experimental design

A. It would be thought-provoking that the authors could discuss the integration of different types of medical data and serological clinical data to enhance the diagnostic accuracy and robustness of the models. Authors are encouraged to cite paper: Lei, R., & Mohan, C. (2020). Immunological Biomarkers of COVID-19. Critical reviews in immunology, 40(6), 497–512. https://doi.org/10.1615/CritRevImmunol.2020035652

B. The review paper is quite lengthy. Please improve the readability of the manuscript, such as by breaking down complex concepts or using more visual aids, could make it more accessible to a broader audience
C. I could not find anywhere stating until when this comprehensive study stopped. Since the field of DL is rapidly evolving, please include a statement in abstract.
D. While the manuscript discusses some limitations and challenges, it could provide a more detailed analysis of these aspects. Please categorize table 10 as Model, Limitation, Potential solutions.
E. Table 11 lists the abbreviations. Please confirm if this is a comprehensive list. Also, it would be appropriate to prepare the list of full names but not place it in the main text.
F. All figures and tables should have a paragraph of legend to convey the message so the reader does not have to go back to main text to understand.
G. I have a difficulty in understanding Figure 12. Please explain.
H. The reference format throughout the context is not consistent. Please correct.

Validity of the findings

No comment

Reviewer 2 ·

Basic reporting

The authors presented a better work entitled, Challenges issues and future recommendations of deep learning techniques for SARS-CoV-2 detection utilising X-ray and CT images: A comprehensive review.
The authors summarize current diagnostic methods heavily rely on imaging techniques like CT scans and X-rays and thoroughly investigate the SARS-CoV-2 study’s classification system, prevailing architecture, structures, and dataset utilization. This encompasses a succinct overview of data attributes, challenges, performance metrics, output evaluations, and prominent recent research citations, shedding light on the foundational equations that underlie deep learning algorithms.

However, authors are recommended to consider the following,
1.A thorough proofread is recommended.
2.A few of the figures can be enhanced to make them more readable.
3.The resolution of some pictures can be increased.
4.Although deep learning has made great progress in medical imaging, sequencing-based COVID-19 detection technology has made important contributions to the prevention, diagnosis, and treatment of COVID-19. The author needs to take a more dialectical view of this issue.
5.This review provides an excessive and unnecessary description of the process of citing its own articles, elaborating on aspects of the field, and the structure of the articles. It is hoped that a more objective and concise introduction of its features can be presented, with a focus on the important application of Deep Learning Techniques in the detection of SARS-CoV-2 in X-ray and CT images.

Experimental design

.

Validity of the findings

.

Reviewer 3 ·

Basic reporting

The manuscript is well-structured, following a clear and logical progression from introduction to conclusions. The language is professional and mostly clear, making it accessible to its intended audience of researchers, healthcare professionals, and policy makers.Seemingly, the literature review is comprehensive, demonstrating a deep understanding of the field and identifying both the strengths and limitations of current studies.

Experimental design

The focus on deep learning techniques for SARS-CoV-2 detection using X-ray and CT images is highly relevant and timely. Seemingly, this manuscript has been well-written and successfully outlines the aims and scope, justifying its significance amidst the pandemic.

Validity of the findings

The conclusion effectively ties the research findings back to the initial research question, providing a clear overview of the study's contributions and potential impact.

The findings are well-supported by the data presented, particularly the performance evaluation of different deep learning models. However, it would be beneficial to discuss the external validity of these models more extensively, considering different demographics and healthcare settings.

Additional comments

The manuscript aims to provide a meticulous and comprehensive review of imaging-based SARS-CoV-2 diagnosis using deep learning techniques. The literature represents a significant contribution to the field of medical imaging and AI research. Seemingly, the manuscript is well-structured and falls within the scope of the journal. However, the following issues need further addressing.

1. The findings are well-supported by the data presented, particularly the performance evaluation of different deep learning models. However, it would be beneficial to discuss the external validity of these models more extensively, considering different demographics and healthcare setting.

2. The manuscript could benefit from a more detailed discussion on the ethical considerations and privacy concerns associated with using patient images for AI training.

3. Add suggestions for future research directions, particularly the emphasis on cross-disciplinary collaboration and the development of more accessible and generalizable models.

---

## Round 0.2 · Minor Revisions

There are some minor concerns that need to be addressed. The authors should provide point-to-point responses to address all the concerns and provide a revised manuscript with the revised parts being marked in different color.

Reviewer 1 ·

Basic reporting

After reviewing the revised paper, the paper has significantly improved the writing and description of contents. No more comment to this section.

Experimental design

Authors also responded my questions point by point and provided clear explanation. The article has reorganized the structure and removed parts of the unnecessary contents, making the article more rigorous and aligned with the topic. No more comment to this section.

Validity of the findings

This is a comprehensive review paper so the novelty is limited. But the paper did provide their expert opinion in the development trajectory of AI application in SARS-Cov-2 diagnostics and management. The conclusion is clear and contextual.

Additional comments

No comment

Reviewer 2 ·

Basic reporting

In my opinion, the authors have effectively responded to all the queries and concerns raised during the previous review round. They have made substantial revisions and clarifications that significantly enhance the clarity and quality of the paper. The revised manuscript addresses all relevant points, demonstrating a thorough understanding of the subject matter. Given these improvements, I believe the paper is now in an appropriate condition for acceptance in its current form. It contributes valuable insights to the field and aligns well with the journal's standards.

Experimental design

In my opinion, the authors have effectively responded to all the queries and concerns raised during the previous review round. They have made substantial revisions and clarifications that significantly enhance the clarity and quality of the paper. The revised manuscript addresses all relevant points, demonstrating a thorough understanding of the subject matter. Given these improvements, I believe the paper is now in an appropriate condition for acceptance in its current form. It contributes valuable insights to the field and aligns well with the journal's standards.

Validity of the findings

In my opinion, the authors have effectively responded to all the queries and concerns raised during the previous review round. They have made substantial revisions and clarifications that significantly enhance the clarity and quality of the paper. The revised manuscript addresses all relevant points, demonstrating a thorough understanding of the subject matter. Given these improvements, I believe the paper is now in an appropriate condition for acceptance in its current form. It contributes valuable insights to the field and aligns well with the journal's standards.

Additional comments

In my opinion, the authors have effectively responded to all the queries and concerns raised during the previous review round. They have made substantial revisions and clarifications that significantly enhance the clarity and quality of the paper. The revised manuscript addresses all relevant points, demonstrating a thorough understanding of the subject matter. Given these improvements, I believe the paper is now in an appropriate condition for acceptance in its current form. It contributes valuable insights to the field and aligns well with the journal's standards.

---

## Round 0.3 · accepted · Accept

Authors have addressed the remaining minor concerns and I recommend accepting this manuscript.